# High-fidelity, efficient, and reversible labeling of endogenous proteins using CRISPR-based designer exon insertion

**Haining Zhong[1]\*, Cesar C Ceballos[1], Crystian I Massengill[1], Michael A Muniak[1], Lei Ma[1], Maozhen Qin[1], Stefanie Kaech Petrie[2], Tianyi Mao[1]**

[1]Vollum Institute, Oregon Health & Science University, Portland, United States; [2]Department of Neurology, Oregon Health & Science University, Portland, United States

**Abstract** Precise and efficient insertion of large DNA fragments into somatic cells using gene editing technologies to label or modify endogenous proteins remains challenging. Non-specific insertions/deletions (INDELs) resulting from the non-homologous end joining pathway make the process error-prone. Further, the insert is not readily removable. Here, we describe a method called *CRISP*R-mediated *i*nsertion of *e*xon (CRISPIE) that can precisely and reversibly label endogenous proteins using CRISPR/Cas9-based editing. CRISPIE inserts a designer donor module, which consists of an exon encoding the protein sequence flanked by intron sequences, into an intronic location in the target gene. INDELs at the insertion junction will be spliced out, leaving mRNAs nearly error-free. We used CRISPIE to fluorescently label endogenous proteins in mammalian neurons in vivo with previously unachieved efficiency. We demonstrate that this method is broadly applicable, and that the insert can be readily removed later. CRISPIE permits protein sequence insertion with high fidelity, efficiency, and flexibility.

**\*For correspondence:** zhong@ohsu.edu

## Introduction

The CRISPR/Cas9 technology has revolutionized genomic editing (*Cong et al., 2013*; *Doudna, 2020*; *Heidenreich and Zhang, 2016*; *Jinek et al., 2012*; *Mali et al., 2013*). However, one of the major unresolved challenges is to use CRISPR-based technologies to precisely and efficiently knock in large DNA fragments in somatic cells of living animals (i.e., without generating transgenic animals) to label or modify endogenous proteins. Such technology holds promise for both studying biological mechanisms and gene therapy.

Toward this goal, several studies have developed CRISPR/Cas9-based approaches to insert fluorescent protein (FP)-encoding sequences to label and visualize endogenous proteins in somatic cells, such as postmitotic neurons (*Artegiani et al., 2020*; *Gao et al., 2019*; *Mikuni et al., 2016*; *Nakade et al., 2014*; *Nishiyama et al., 2017*; *Schmid-Burgk et al., 2016*; *Suzuki et al., 2016*; *Uemura et al., 2016*; *Willems et al., 2020*). Fluorescent visualization of proteins in living cells is essential for the mechanistic dissection of cellular and organismic processes because many functions of cells and organisms are established and manifested by their constituent proteins. However, conventional methods have fallen short in this regard: immunolabeling is typically incompatible with live imaging, and it usually does not permit sparse labeling for cell-specific contrast in tissues. FP tagging typically involves overexpression, which may result in undesired off-target effects. Knock-ins of FP tags are costly and time-consuming in mammalian species, and, with the exception of certain complex schemes (e.g., *Fortin et al., 2014*; *He and Huang, 2018*), are typically associated with poor contrast due to global expression of the FP-tagged protein of interest (e.g., *Herzog et al., 2011*). In

contrast, if successful, CRISPR-based FP labeling of endogenous proteins in somatic cells may overcome these limitations and remove a major bottleneck in studying biological mechanisms.

Unfortunately, current CRISPR-based somatic protein labeling technologies are faced with major limitations. First, the in vivo labeling efficiency is moderate, especially for the large DNA insertions required for FP labeling (~15% or less) (*Mikuni et al., 2016*; *Nishiyama et al., 2017*; *Suzuki et al., 2016*; *Uemura et al., 2016*). Second, and more importantly, these methods, which target protein coding sequences or their immediately adjacent exonic sequences, are error-prone. The precise insertion of short (<50 bp) DNA sequences has started to become possible via prime editing (*Anzalone et al., 2019*), which only nicks the DNA; however, the insertion of larger DNA fragments still requires the generation of double-stranded DNA breaks (DSBs). At the edited loci involving DSBs, including those loci where the DNA insertion does not occur, the efficient non-homologous end joining (NHEJ) pathway often results in unwanted insertions/deletions (INDELs) that cause mutations and/or frame shifts (*Cong et al., 2013*; *Doudna, 2020*; *Heidenreich and Zhang, 2016*; *Mali et al., 2013*) (see also Figure 2 and *Figure 1—figure supplement 1*). This is the case regardless of whether NHEJ or homologous-dependent repair (HDR) is engaged for the insertion process (*Roberts et al., 2017*). Finally, it has been increasingly recognized that reversible DNA editing may allow for great flexibility in both research and gene therapy (*Nakamura et al., 2019*); however, the inserted labels are not readily removable at a later time using existing approaches.

Here, we present a strategy, called *CRISP*R-mediated *i*nsertion of *e*xon (CRISPIE), which allows for the nearly error-free insertion of FP sequences with high efficiency. Instead of targeting gene exons, CRISPIE targets introns and inserts a designer donor module, which includes an exon encoding the desired protein sequence and the surrounding intronic sequences. INDELs occurring at the insertion junction within the intronic region of DNA will be spliced out (*Figure 1A*), resulting in very low error rates at the mRNA level (>98% correct). CRISPIE is flexible and broadly compatible with: (1) both N- and C-terminal labeling, (2) proteins with diverse structures and functions, including pre- and post-synaptic proteins and cytoskeletal proteins, (3) all major transfection methods, (4) FPs with diverse colors, and (5) multiple animal species. In part because introns offer ample editing sites to choose from, and because INDELs at the DNA level do not affect the success of editing, a high labeling efficiency (up to 43%) was achieved in cortical neurons of living mice. Importantly, CRISPIE-mediated DNA insertions are erasable. By flanking the donor module with additional designer CRISPR editing sites in the intronic region, the inserted DNA fragment can be erased at a later time. CRISPIE may allow for the routine labeling of proteins at endogenous levels and can be expanded to the insertion of other genetically encoded functional sequences to manipulate protein function.

## Results

### Design and demonstration of CRISPIE

To achieve high-fidelity FP labeling, we developed the CRISPIE method, which inserts a designer exon module that consists of an exogenous exon encoding the FP flanked by intronic sequences, including the splicing acceptor and donor sites, into the intronic region of the target gene via the NHEJ pathway (*Figure 1A*, *Figure 1—figure supplement 1*). NHEJ exhibits a higher editing efficiency than HDR in postmitotic somatic cells, such as neurons (*Heidenreich and Zhang, 2016*; *Heyer et al., 2010*; *Mao et al., 2008*; *Saleh-Gohari and Helleday, 2004*). Because introns are spliced out of mRNAs, they are often poorly conserved and tolerate stochastic mutations better than exons (e.g., see *Figure 1—figure supplement 2*). INDELs at the joint junction of the insertion therefore will not result in mutations in the mRNA and thus the encoded protein under most circumstances (*Figure 1A*). In addition, NHEJ-mediated donor DNA insertion can result in unwanted integration with inverted orientation, and INDELs can occur at loci where the donor fails to insert. However, neither of these events will produce disrupted mRNAs when using CRISPIE (*Figure 1A*). Conceptually, unlike conventional exonic targeting, CRISPIE only results in either wild-type or successfully labeled mRNAs. For convenience, after a gene is labeled by CRISPIE, we say it has been 'CRISPIEd'.

As a proof of principle, we first labeled human β-actin (ACTB) at its N-terminus with monomeric EGFP (mEGFP) in human U2OS cells by co-transfecting a plasmid that expresses a single guide RNA (sgRNA) targeting the *ACTB* gene and *Streptococcus pyogenes* Cas9 (SpCas9), a plasmid carrying

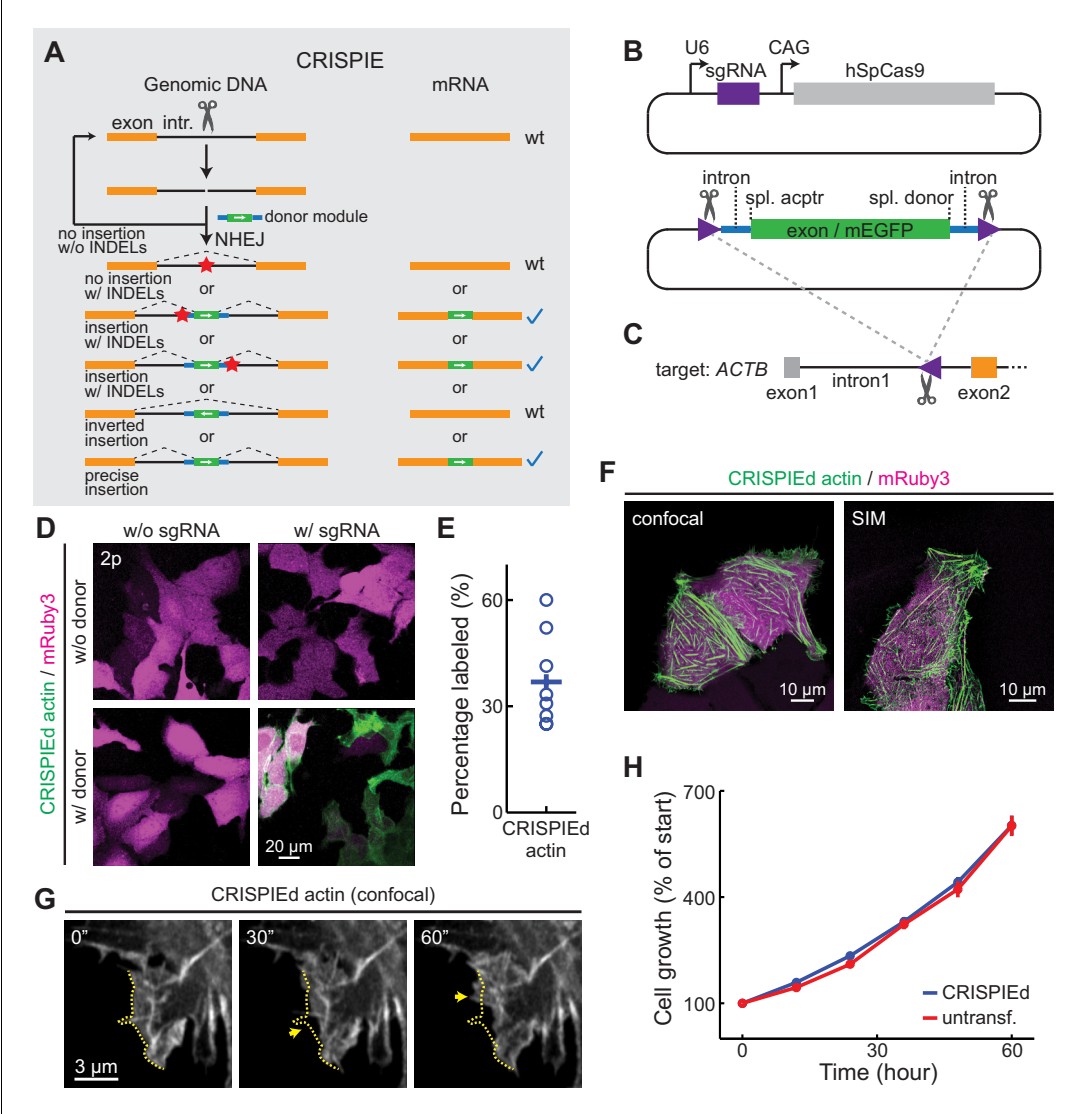

**Figure 1.** The CRISPR-mediated insertion of exon (CRISPIE) strategy and its application to label human β-actin. (**A**) The conceptual design of the CRISPIE strategy showing that, although insertions/deletions (INDELs) and inverted insertion events may occur at the DNA level during editing, only wild-type mRNAs or mRNAs with the desired precise insertion are produced. Orange boxes: endogenous exon sequences; black lines: endogenous intron sequences; blue lines and green boxes: intron and exon sequences, respectively, of the designer donor module with white arrows showing the correct orientation of the reading frame; red stars: INDELs. (**B**) Schematics of the single guide RNA/*Streptococcus pyogenes* Cas9 (sgRNA/SpCas9) plasmid (upper) and the donor plasmid (lower) that are used for panels **D–G**. Purple triangles: the sgRNA-targeted location and orientation, spl.: splicing; acptr: acceptor. See *Figure 1—figure supplement 3* for design details. (**C**) Schematic of the targeted intron of *ACTB* with the purple triangle showing the sgRNA targeting site and orientation. Orange and gray boxes: exonic sequences that do and do not encode protein sequences, respectively. (**D**) Representative two-photon (2 p) images of live U2OS cells with the indicated transfection. Note that the mRuby3 (magenta) expression levels were highly variable across cells due to the variability in plasmid transfection; however, the green label intensities were comparable across cells, as expected for the expression from an endogenous locus. (**E**) Labeling efficiency (green cell counts over red cell counts) for ACTB in U2OS cells. n = 8 field of views (FOVs), two independent transfections. (**F**) Representative confocal (Airyscan) and super-resolution structured illumination microscopy (SIM) images of live U2OS cells expressing CRISPIEd β-actin. (**G**) Representative time-lapse confocal images of live U2OS cells showing the dynamics of actin ruffles (arrows). The dashed yellow line outlines the cell morphology at time zero. (**H**) Cell growth curves of β-actin-CRISPIEd cells and untransfected cells. n = 6 FOVs.

The online version of this article includes the following source data and figure supplement(s) for figure 1:

**Source data 1.** Numeric data for *Figure 1E*.

**Figure supplement 1.** DNA insertion targeting coding sequences result in many possible modes of unwanted mutations.

**Figure supplement 2.** Introns are much less conserved than exons.

**Figure supplement 3.** Schematic of targeting sites and donor designs.

*Figure 1 continued on next page*

*Figure 1 continued*

**Figure supplement 4.** Cell growth curve of *ACTB* CRISPIEd U2OS cells inserted at a intronic location different from *Figure 1H* and *TUBA1B* CRISPIEd U2OS cells.

the donor module (*Figure 1B* and *Figure 1—figure supplement 3B1*), and a transfection marker plasmid that expresses mRuby3. The sgRNA/SpCas9 complex cuts an editing site at the first intron of *ACTB* (*Figure 1C*), as selected using the BROAD Institute sgRNA designer (*Doench et al., 2016*), and releases the donor module from the plasmid by cutting sgRNA-targeted sequences flanking the module (*Figure 1B*, lower panel). The released module is then inserted into the cut site at the *ACTB* intron via the NHEJ pathway. The module-flanking sgRNA-targeted sequences are in the reverse orientation to facilitate the insertion of the module in the forward orientation (*Suzuki et al., 2016*). The donor module includes an exon in the appropriate translational phase (0–0 for intron 1 of *ACTB*) encoding mEGFP and peptide linkers (*Figure 1—figure supplement 3B1*). At the exon-intron junctions, ~100 bp of intronic sequence and ~10 bp of adjacent exonic sequence were taken from an obligatory intron (i.e., no reported alternative splicing events) and its adjacent exons of mouse CaMKIIα (*Camk2a*), a highly expressed protein in the brain, to include the splicing donor or acceptor site (*Wang and Burge, 2008*). mEGFP labeling was observed at 3–4 days post-transfection (*Figure 1D*) and required the presence of both sgRNA/SpCas9 and the donor module. Over 35% (37 ± 5%) of transfected cells, as identified by the expression of mRuby3 (red), were mEGFP (green)-labeled (*Figure 1E*). This likely represents a lower bound of labeling efficiency, since the co-transfection rate of all three plasmids is unlikely to reach 100%. Confocal and super-resolution structured illumination microscopy (SIM) revealed that the mEGFP signals exhibited the characteristic distribution of actin protein (*Fischer et al., 1998*; *Planchon et al., 2011*; *Figure 1F*). Time-lapse confocal microscopy revealed dynamic actin ruffles at the edges of cells (*Figure 1G*), which is consistent with the current knowledge of actin dynamics (*Fischer et al., 1998*; *Planchon et al., 2011*). Furthermore, cells with CRISPIEd actin grew at a rate indistinguishable from that of unlabeled cells in the same dish (*Figure 1H* and *Figure 1—figure supplement 4*). These results are consistent with the predictions for a functional, FP-labeled endogenous actin protein.

## CRISPIE-mediated protein insertion is nearly error-free at the mRNA level

To characterize the insertion events and INDEL rates at the insertion site, we sorted transfected cells into red-only (transfected, but not labeled) and green (transfected and labeled) cells. The actin genomic DNA and mRNA at the editing locus were analyzed using PCR and RT-PCR, respectively (*Figure 2A and B*; see *Figure 2—figure supplement 1* and *Supplementary files 1* and *2* for primers, and anticipated PCR products and sizes). As expected, bands corresponding to the successful insertion (i.e., insertion in the forward orientation) at the targeted site are observed in green-labeled cells for both the genomic DNA and the mRNA (*Figure 2A*). Systematic analyses of green-labeled cells detected all possible editing events at the genomic DNA level, including no label insertion, forward insertion, and inverted insertion (*Figure 2B*), indicating that insertion events were typically non-homologous (i.e., both chromosomes differentially edited). Had the coding sequence been targeted, an inverted label insertion would inevitably cause disruptive mutations. However, under our conditions, because the donor only contained the splicing signals for exon inclusion in the forward orientation, only wild-type and forwardly inserted mRNAs were detected (*Figure 2B*), demonstrating the advantage of CRISPIE.

To evaluate how INDELs may affect a targeted gene, the PCR amplicons corresponding to no label insertion events in untransfected, red-only, and green cells, as well as those corresponding to forward insertion events in green cells were subjected to next-generation sequencing (NGS) (*Figure 2C and D*; sequenced amplicons correspond to gel bands in *Figure 2A and B*, as indicated by Roman numerals). Remarkably, at the genomic DNA level in red-only cells, over 70% (834/1181 reads) of PCR amplicon sequences contained INDELs (*Figure 2C*, and second column of *Figure 2D*), confirming the predicted high rate of INDELs during gene editing. In contrast, at the mRNA level in these cells, 99.7% (2930/2936 reads) of the sequences did not contain INDELs ($p<0.001$, *cf.* genomic DNA) (*Figure 2D*, second column), which was no different from control, untransfected cells (99.7%,

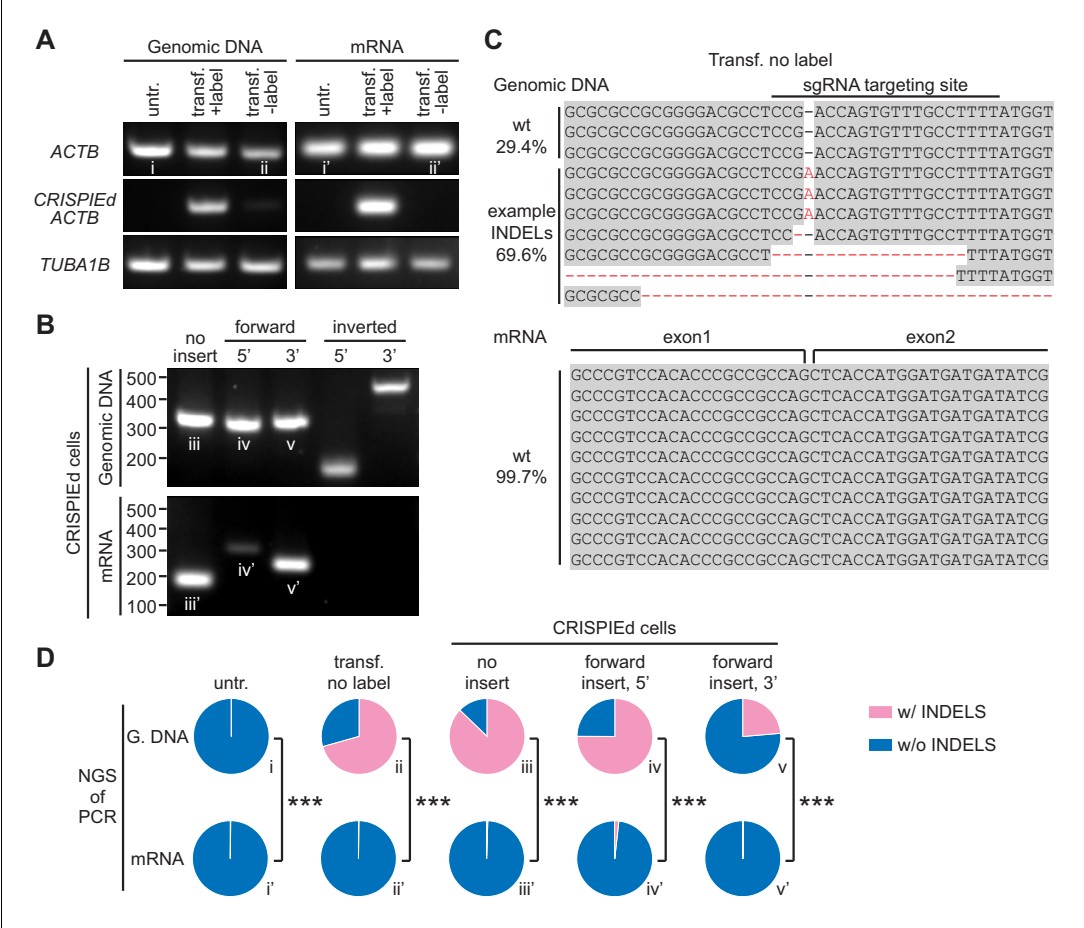

**Figure 2.** CRISPR-mediated insertion of exon (CRISPIE)-mediated β-actin labeling is resistant to inverted insertion events and insertions/deletions (INDELs). (**A**) Gel images of PCR analyses for genomic DNA and RT-PCR analyses for mRNA for untransfected (untr.) and transfected U2OS cells with (transf. +label) and without (transf. -label) successful actin labeling. As detailed in *Figure 2—figure supplement 1*, specific primers were used for the targeting site at the *ACTB* gene and mRNA with and without the desired label insertion, and for a control site (*TUBA1B*) that is not edited in this experiment. (**B**) Gel images of PCR and RT-PCR analyses for the genomic DNA and mRNA, respectively, of successfully labeled cells to detect non-labeled events, and both 5' and 3' junctions of forward and inverted label insertion events. (**C**) Representative next-generation sequencing (NGS) results of the genomic DNA and mRNA prepared from transfected U2OS cells without successful actin labeling corresponding to bands ii and ii', respectively. INDELs are marked in red. (**D**) Pie graphs of the relative proportions of INDEL events based on NGS results of the PCR/RT-PCR products from panels **A** and **B**, as marked by the Roman numerals. Statistical tests were performed using a $\chi^2$ test. From top to bottom, n = 54216, 1181, 1589, 2767, and 6036 for genomic DNA, and 5771, 2936, 4166, 3613, and 5949 for mRNA.

The online version of this article includes the following source data and figure supplement(s) for figure 2:

**Source data 1.** Numeric data for *Figure 2D*.

**Figure supplement 1.** Schematic of PCR and RT-PCR products.

**Figure supplement 2.** Next-generation sequencing (NGS) analysis of the genomic DNA of different insertion events.

5761/5771 reads) (*Figure 2D*, first column). Similarly, undesired INDELs were frequently detected in all sequenced amplicons from green cells at the genomic DNA level (*Figure 2D*). Similar high rates of INDELs were also observed at the genomic DNA level in other insertion locations of *ACTB* and in another gene (*TUBA1B*) across two different cell lines (*Figure 2—figure supplement 2*). However, few errors were detected at the mRNA level (INDELs 24–87% for DNA vs. 0.3–2% for mRNA; *Figure 2D*). These numbers correspond to correct-to-mutation ratios of 50- to 300-fold for mRNA, which are around two orders of magnitude higher than those for DNA. These results demonstrate that, although inverted insertion and INDEL events may occur at the DNA level—which would have caused mutations or frame shifts when using existing strategies to target coding sequences—CRISPIE is resistant to these disruptive events at the mRNA level.

## CRISPIE can be optimized to achieve high labeling efficiency

For insertion strategies that target coding sequences, the options for targeting sites are limited, and not all potential targeting sites allow for high-efficiency editing (*Doench et al., 2014*). Furthermore, as discussed above, INDELs cause mutations or frame shifts, thereby reducing the likelihood of successful labeling. In contrast, CRISPIE is insensitive to INDELs, and the lengths of introns usually offer ample choices for potential targeting sites. CRISPIE may therefore have the potential to achieve higher labeling efficiency than strategies that target coding sequences. To test this, we first asked whether different intronic insertion sites may exhibit different labeling efficiencies. Five different editing sites (i1 to i5 in *Figure 3A*) were selected using the BROAD Institute sgRNA designer (with Azimuth v2.0 scores of 0.57–0.69) (*Doench et al., 2016*). By using a generic donor that was released from the plasmid by a designed sgRNA (called DRS-2; see *Figure 1—figure supplement 3B7* for donor plasmid design) (*Gao et al., 2019*), we found that all five sites permitted the successful labeling of ACTB; however, the labeling efficiencies varied significantly (*Figure 3B*), with the highest labeling efficiency corresponding to around 20-fold of the lowest.

We also examined FP labeling that directly targets the coding sequence for comparison. There are only two possible SpCas9 editing sites around the start codon of *ACTB* (referred to as e1 and e2; *Figure 3A*). Despite using their optimal insertion donors (for both, see *Figure 1—figure supplement 3B6*), both exonic sites exhibited significantly lower labeling efficiencies compared to the optimized CRISPIE editing (greater than fivefold; *Figure 3C*). Between the e1 and e2 sites, the one with the higher Azimuth score was in fact less efficient (Azimuth scores = 0.55 and 0.62, respectively), indicating that the labeling efficiency is not yet entirely predictable. Nevertheless, CRISPIE allows for systematic optimization by screening for highly efficient editing sites and, when optimized, can achieve higher labeling efficiency than exonic editing.

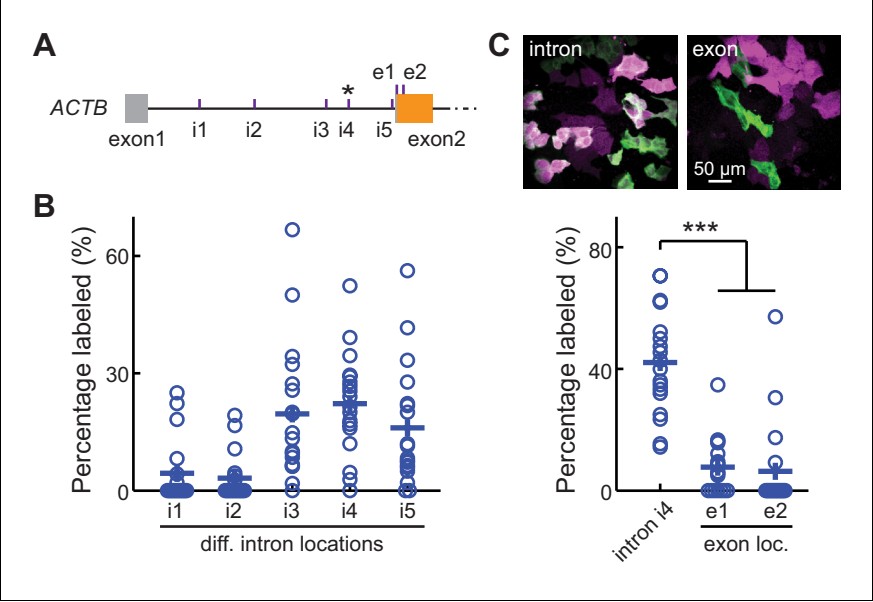

**Figure 3.** CRISPR-mediated insertion of exon (CRISPIE) is optimizable and achieves higher efficiency than exon labeling. (**A**) Schematic positions of the tested editing sites in the intron 1 and exon 2 of *ACTB* that are targeted in panels **B** and **C**. The asterisk marks the site used for *ACTB* in subsequent experiments. (**B**) Comparison of the editing efficiency of five different intronic locations. For comparison under identical conditions, a generic donor excised using an independent single guide RNA (sgRNA) (DRS-2) was used (*Figure 1—figure supplement 3B7*). n = 18 FOVs (field of views) from two independent transfections. (**C**) Representative images and quantification of successful editing rates at the intronic location i4 vs. the only two possible exonic locations for N-terminally labeled *ACTB*, each using their specifically designed donors. n = 18 FOVs from two independent transfections. The online version of this article includes the following source data for figure 3:

**Source data 1.** Numeric data for *Figure 3B and C*.

## CRISPIE is broadly applicable in dividing cells

The CRISPIE method should be broadly applicable for use with different FPs or with other functional domains, different protein targets, and different animal species. To test different FP labels for potential multiplex imaging studies, we successfully labeled ACTB in U2OS cells using cyan (mTurquoise2), green-yellow (mNeonGreen), and red (mRuby3) FPs (*Figure 4A*, donor plasmids illustrated in *Figure 1—figure supplement 3B2-4*). Notably, when both mTurquoise2 and mNeonGreen donors were included simultaneously, double-labeled cells could be found, which appeared to be capable of dividing (*Figure 4—figure supplement 1*), indicating that FP labeling does not affect cell viability in diploid CRISPIEd cells. To test the applicability of CRISPIE to different protein targets, we labeled several additional important subcellular structures in U2OS cells, namely microtubules, mitochondria, focal adhesion complexes, and the endoplasmic reticulum, by labeling tubulin alpha 1B (TUBA1B), translocase of outer mitochondrial membrane 20 (TOMM20), vinculin (VCL), and calreticulin (CALR), respectively (*Figure 4B* and *Figure 1—figure supplement 3A and B*; *Lam et al., 2012*; *Rizzo et al., 2009*; *Roberts et al., 2017*; *Shroff et al., 2008*). In a growth test, cells with CRISPIEd *TUBA1B* grew at a rate indistinguishable from that of unlabeled cells in the same dish (*Figure 1—figure supplement 4*), suggesting that such labeled TUBA1B is functional. Notably, the *TOMM20* and *VCL* genes were labeled at their respective last introns (*Figure 1—figure supplement 3A*) because their encoded proteins are best tagged at their C-termini (*Rizzo et al., 2009*; *Roberts et al., 2017*; *Shroff et al., 2008*). This demonstrates that CRISPIE is also readily compatible with C-terminal labeling. For some genes, such as *VCL*, there are coding sequences of significant lengths present at the

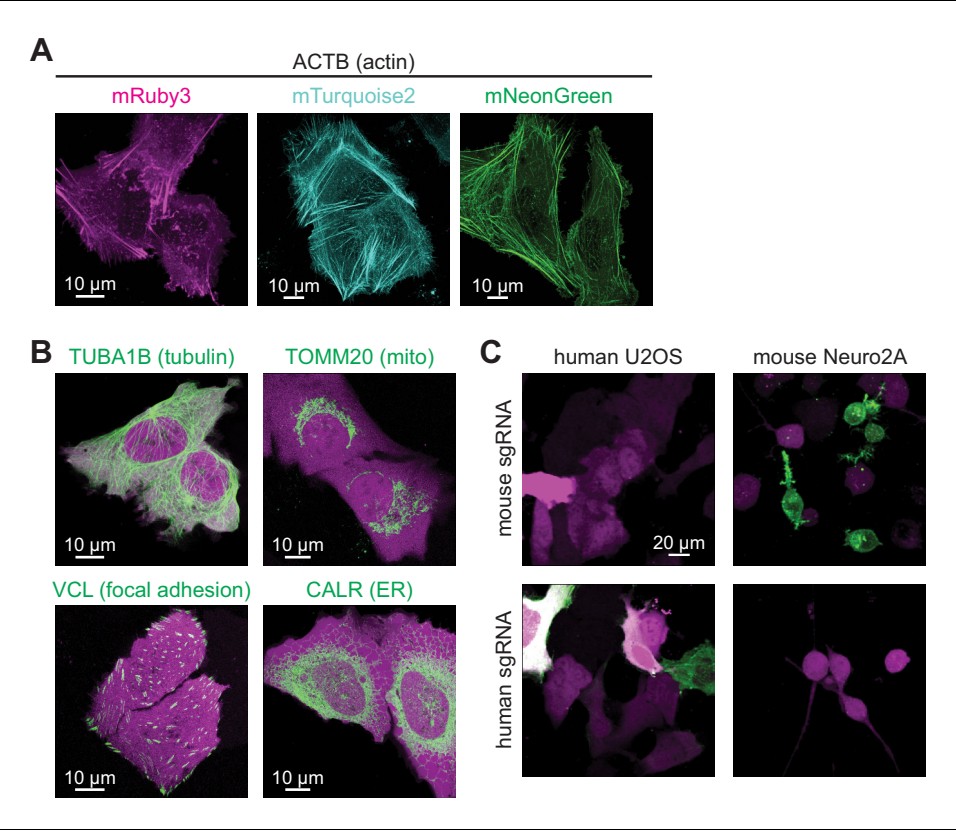

**Figure 4.** CRISPR-mediated insertion of exon (CRISPIE) is applicable to labeling with different colors, different proteins, and different animal species. (**A**) Representative images of ACTB labeled with each of the three indicated fluorescent proteins (FPs) of different colors in U2OS cells. (**B**) Representative images of four different proteins, as indicated, labeled with monomeric EGFP (mEGFP) in U2OS cells. (**C**) Representative images of human and mouse β-actin labeling in human U2OS cells and mouse Neuro 2A cells, respectively, under the indicated conditions. The online version of this article includes the following figure supplement(s) for figure 4:

**Figure supplement 1.** Dual color labeling within the same cells.

last exon, downstream of the possible CRISPIE labeling site. To prevent potential disruption of protein function caused by an insertion in the middle of the protein, the coding sequence of the last exon of *VCL* was included in the insertion donor sequence, and a stop codon was introduced at the end of the mEGFP coding sequence (*Figure 1—figure supplement 3B*). To test the applicability in different animal species, we further labeled β-actin (Actb) in mouse Neuro 2A cells. A mouse-specific sgRNA was used because introns are degenerative across species (e.g., see *Figure 1—figure supplement 2*). The labeling was successful and was dependent on the correct combination of the species and the corresponding sgRNA (*Figure 4C*). CRISPIE labeling was also successful in rat neurons

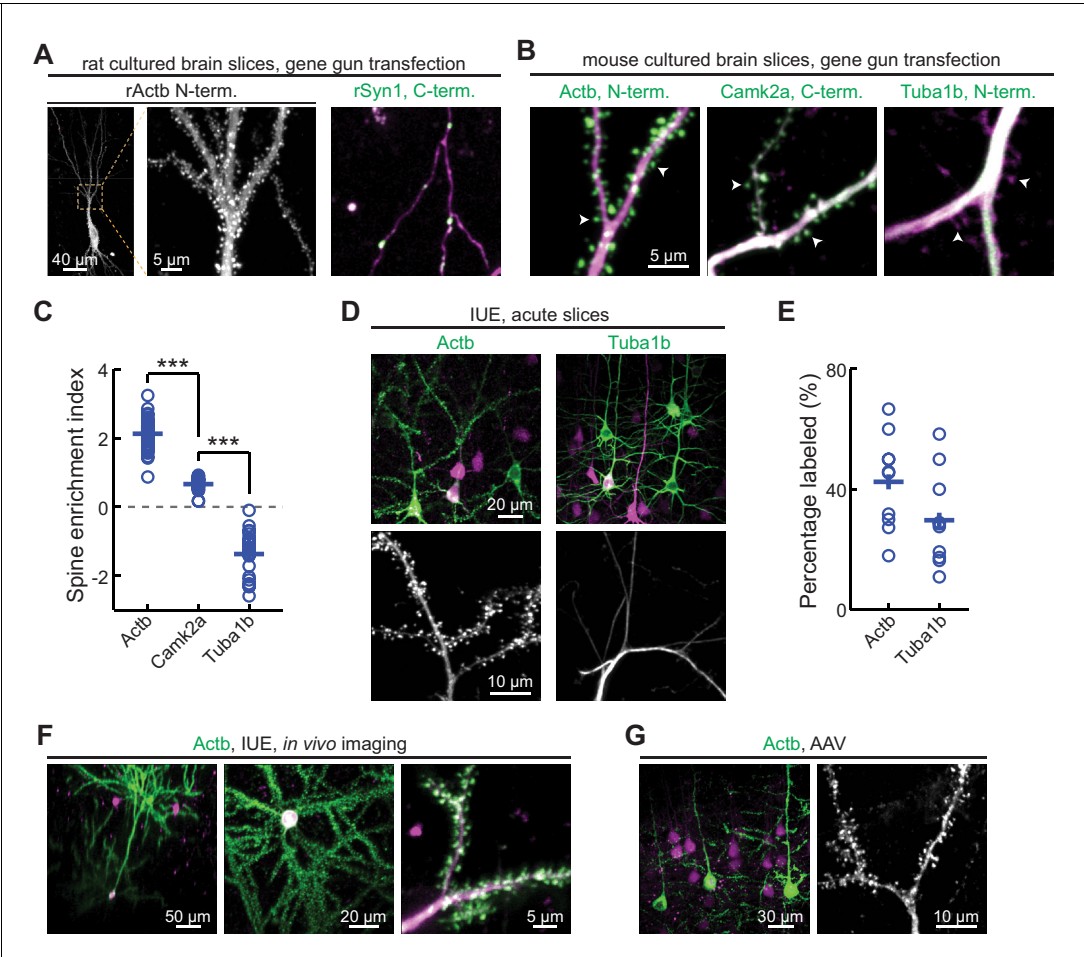

**Figure 5.** CRISPR-mediated insertion of exon (CRISPIE) is applicable to different proteins in postmitotic neurons using various transfection methods. (**A**) Representative images of the indicated proteins labeled in rat cultured hippocampal slices transfected using the biolistic method. (**B**) Representative images of the indicated proteins labeled in cultured hippocampal slices from *SpCas9* ± mice transfected using the biolistic method. Arrowheads indicate spines where each of the three proteins is differentially enriched or excluded. (**C**) Spine enrichment index (SEI) of the three proteins in panel **B**. From left to right, n (spines/neurons)=55/5, 26/3, and 22/2. (**D, E**) Representative images (**D**) and labeling efficiency quantifications (**E**) of β-actin or α-tubulin 1B-labeled neurons that are labeled in vivo via in utero electroporation (IUE) in the mouse somatosensory cortex and imaged in acute brain slices. n = 10 FOVs (field of views) from three mice for the quantification of both proteins. (**F**) Representative side view (x-z, left), top view (x-y, middle), and zoomed-in (x-y, right) images of labeled Actb (green) in cortical neurons that were transfected via IUE and were imaged in living mice via a cranial window. (**G**) Representative images of β-actin-CRISPIEd neurons in acute brain slices labeled via adeno-associated virus (AAV) injections into the cortex of *SpCas9* ± mice.

The online version of this article includes the following source data and figure supplement(s) for figure 5:

**Source data 1.** Numeric data for *Figure 5C and E*.

**Figure supplement 1.** The CRISPR-mediated insertion of exon (CRISPIE) method is suitable for labeling neurons in cultured hippocampal slices transfected using the biolistic (gene gun) method.

**Figure supplement 2.** Neurons with CRISPIEd rActb exhibited normal miniature excitatory postsynaptic current (mEPSC) frequencies and amplitudes.

**Figure supplement 3.** CRISPIEd Actb transfected using in utero electroporation is suitable for in vivo imaging.

(*Figure 5A*, see below). Overall, these results indicate that CRISPIE is versatile and readily applicable to many different FP labels, protein species, and animal species, and can be used to tag proteins at their N- and C-termini.

## CRISPIE is broadly applicable to postmitotic neurons using various transfection methods in vitro and in vivo

To determine whether CRISPIE can be used in postmitotic cells, such as neurons, and whether it is compatible with different transfection methods, we applied it to label proteins in neurons in cultured slices and in vivo using three common transfection methods. First, we used the biolistic method (i.e., 'gene gun') to deliver the sgRNA and donor for rat β-actin (rActb), and the cytosolic marker mRuby3, to cultured rat hippocampal slices. Notably, although the biolistic transfection method is widely used in studying the cellular functions of neurons, CRISPR-based labeling has not yet been demonstrated using this method, likely due in part to the relatively low labeling efficiencies. For this same reason, high-contrast double labeling—for example, the target protein together with a cell morphology marker—in brain tissue remains challenging. In our experiments, 6–45 neurons were transfected per brain slice, as determined by their red mRuby3 fluorescence. Among them, on average 15% (15 ± 4%, n = 6 slices) exhibited successful labeling of actin (*Figure 5A*). In some slices, as many as six to seven successfully labeled neurons of different neuronal types, including dentate gyrus neurons, CA1 and CA3 pyramidal neurons, and interneurons, could be detected (*Figure 5—figure supplement 1*). Glial cells could also be labeled (*Figure 5—figure supplement 1B*). The labeled neurons exhibited normal miniature excitatory postsynaptic currents (mEPSCs) in frequency and amplitude when compared to adjacent untransfected neurons (*Figure 5—figure supplement 2*). These results indicate that CRISPIE provides a viable option for protein labeling using gene gun in cultured slices. Similar results were obtained in cultured mouse hippocampal slices from an *SpCas9* heterozygous genetic background, which constitutively expresses SpCas9 and thereby simplified the experimental design (*Figure 5B*). Other synaptic proteins, including synapsin (SYN1), tubulin (TUBA1B), and CaMKIIα, were also successfully labeled (*Figure 5A and B*). In many labeled cells, both the mRuby3 cytosolic marker and mEGFP were readily detectible, allowing us to compare the subcellular distribution of three proteins—β-actin, α-tubulin 1B, and CaMKIIα—in neuronal dendrites (*Figure 5C*). By quantifying their distribution between the dendrites and spines using a previously described spine enrichment index (SEI) (*Zhong et al., 2009*), we found that the three proteins exhibited distinct subcellular distributions: actin was highly enriched in dendritic spines, and CaMKIIα was moderately enriched, whereas tubulin was excluded from spines (*Figure 5C*; see **Materials and methods** for quantification details).

To determine whether CRISPIE can be used to label endogenous proteins in vivo, in utero electroporation (IUE) was used to introduce the sgRNA, the donor, and the cytosolic marker mRuby3 into the somatosensory cortex of *SpCas9* mice to label β-actin or tubulin in separate experiments (*Figure 5D*). Among the transfected red cells, the labeling efficiencies were 43 ± 5% for actin and 30 ± 5% for tubulin (*Figure 5E*), both of which are significantly higher than previously reported for CRISPR-based endogenous labeling of any protein in vivo (*Mikuni et al., 2016*; *Nishiyama et al., 2017*; *Suzuki et al., 2016*; *Uemura et al., 2016*). This high level of labeling efficiency allowed us to routinely perform in vivo imaging in the brains of living mice (*Figure 5F* and *Figure 5—figure supplement 3*). Finally, as viruses have become an increasingly prevalent method of transfection, we generated an adeno-associated virus (AAV) harboring the sgRNA targeting β-actin, the donor, and mRuby3 (*Figure 1—figure supplement 3C*). When injected into the cortex of *SpCas9* mice, this virus successfully transduced neurons in vivo and labeled the actin cytoskeleton (*Figure 5G*). Overall, CRISPIE can be readily applied using a variety of widely used methods to transfect neurons both in vitro and in vivo with high efficiency.

## CRISPIE can be reversible

Reversible gene editing may offer significant benefits in research, and in gene therapy in the future. However, existing CRISPR-based methods for endogenous labeling and large-fragment DNA insertions are not readily reversible. CRISPIE allows for the inclusion of functional elements at the flanking intronic regions of the donor module without affecting the mRNA. Taking advantage of this, we included additional designer sgRNA targeting sequences (DRS-1; *Gao et al., 2019*) at both ends of

the module (*Figure 6A*). These added sites allowed for the later excision of the inserted module guided by the DRS-1 sgRNA. To demonstrate this reversibility, we isolated a U2OS cell clone of *CRISPIEd ACTB*. All cells derived from this clone uniformly exhibited green-labeled actin structures (*Figure 6B*, left panel). When DRS-1 sgRNA and SpCas9 were co-transfected with the transfection marker mRuby3, the green label was absent or greatly attenuated in over three quarters (76 ± 6%) of the mRuby3-positive cells at 5 days after transfection (*Figure 6B and C*). As a control, transfection with an sgRNA targeting mouse β-actin did not result in detectible change in the green labeling, indicating that CRISPIE is a readily reversible gene editing approach at the DNA level.

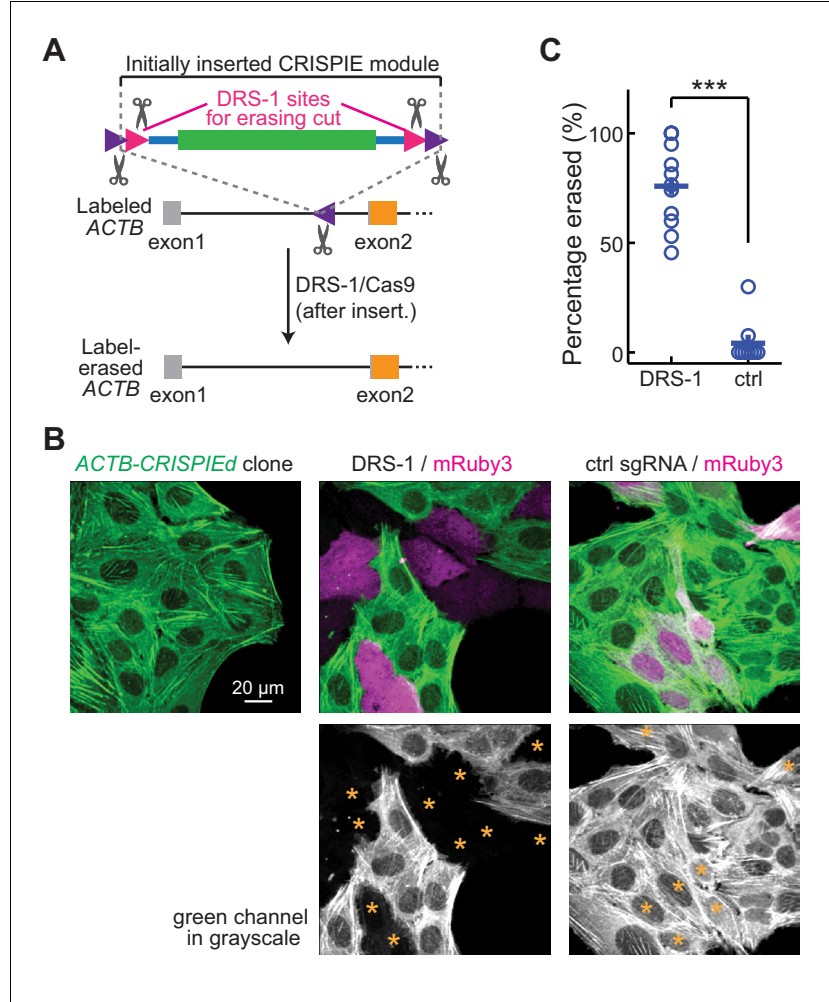

**Figure 6.** Labeling with CRISPR-mediated insertion of exon (CRISPIE) can be erased. (**A**) Schematic of the experimental design for erasing CRISPIE labels. DRS-1 single guide RNA (sgRNA) targeting sequences were used to flank the exon/intron module inside the other two sgRNA sites for excising the donor from the initial insertion. Later introduction of DRS-1 sgRNA and *Streptococcus pyogenes* Cas9 (SpCas9) can excise the inserted CRISPIE module. Note that after the insertion and erasure, both the original sgRNA targeting site and the DRS-1 site will be destroyed. (**B**) Representative images of a β-actin-CRISPIEd U2OS cell clone (left), which was transfected with the transfection marker mRuby3 (magenta in upper panels) and SpCas9 together with DRS-1 (middle), or with a control sgRNA that targeted mouse, but not human β-actin (right). Asterisks in the lower panel mark the transfected (i.e., mRuby-positive) cells. (**C**) Quantification of the erasure efficiency among the transfected cells, as identified by mRuby3 expression. n (field of views [FOVs]/transfections)=11/3 for DRS-1, and 9/3 for control sgRNA. The online version of this article includes the following source data for figure 6:

**Source data 1.** Numeric data for *Figure 6C*.

## Discussion

Here, we present CRISPIE, a reversible CRISPR/Cas9-based approach for inserting large DNA fragments to label or modify endogenous proteins that is essentially impervious to the side effects of INDELs. CRISPIE achieves a high level of precision by inserting a designer exon module flanked by adjacent introns into an intronic site of the target gene. While intronic editing has been used to introduce non-coding DNA tags, microRNAs, gene disruptions, exon replacements, and super-exons (*Bednarski et al., 2016*; *Chen et al., 2018*; *Jarvik et al., 1996*; *Lee et al., 2018*; *Li et al., 2016*; *Miura et al., 2015*), these studies involved cloning of the cell or organism. Although CRISPIE can conceivably be used when cloning of the edited cell or organism is involved, its strength lies in its applicability to somatic cells. Somatic cell editing offers significant advantages compared to cloning-based approaches, including lower cost, faster turnaround time, and potentially controllable alterations in labeling density. In particular, with regard to labeling density, sparse labeling is desired for high-contrast imaging, whereas high-density labeling is needed for biochemical or proteomic analyses. In addition, somatic cell editing will be required under most circumstances when gene editing is applied directly on patients for therapeutic purposes.

As mentioned earlier, several previous studies have labeled endogenous proteins in somatic cells (*Gao et al., 2019*; *Mikuni et al., 2016*; *Nishiyama et al., 2017*; *Schmid-Burgk et al., 2016*; *Suzuki et al., 2016*; *Uemura et al., 2016*; *Willems et al., 2020*). However, these methods are sensitive to INDELs because they directly target exons. Notably, when these methods are applied in vivo, the majority of transfected cells are not labeled (labeling efficiency $\leq$15%), but INDELs can still occur and cause major disruption of the targeted protein in these unlabeled cells. In the (v)SLENDR approach (*Mikuni et al., 2016*; *Nishiyama et al., 2017*), which engages HDR as the insertion mechanism, the sequences within a short range before the start codon or after the stop codon can be used as target sites, thereby alleviating the side effects of INDELs. However, INDELs often include deletions tens of base pairs or longer (e.g., *Figure 2C*). For many genes, INDELs in the 5' and 3' untranslated regions may potentially affect the trafficking, stability, or translational control of the mRNA (*Fortin et al., 2014*; *Goldie and Cairns, 2012*). Furthermore, HDR occurs with low efficiency in non-dividing cells. The vSLENDR approach increases HDR efficiency in postmitotic neurons using high-titer AAV; however, AAV construction and generation add significantly to the cost and time of experiment, and the in vivo labeling efficiency remains low ($\leq$15%). Finally, the recently developed prime editing technique, while exhibiting high precision, can only insert short sequences (<50 bp) and has not been demonstrated in vivo.

Compared to previous methods, CRISPIE is nearly error-free in both labeled cells, and transfected but unlabeled cells. It is broadly compatible in vitro and in vivo with all tested transfection methods, including liposome-based transfection, biolistic transfection, in utero electroporation, and viral infection. CRISPIE exhibits approximately fivefold higher labeling efficiency in side-by-side comparisons with an exon targeting strategy, and achieves up to 2.7-fold higher efficiency than the best in vivo labeling efficiency described in the literature (*Nishiyama et al., 2017*; *Suzuki et al., 2016*). This higher labeling efficiency is significant as it greatly increases the feasibility of in vivo multiplexed imaging of the labeled protein with the cellular context revealed by a cytosolic marker (e.g., *Figure 5F*). It also provides a new starting point for further improving the labeling efficiency toward 100%, which may be required for future therapeutic purposes.

CRISPIE is limited by the availability of introns at or near the desired targeting sites. However, as demonstrated, such sites are available in the genes of a large number of protein species for labeling at either the N- or C-termini. In addition, labeling at an internal site of a gene is feasible as long as the insertion does not disrupt the function of the encoded protein. Many introns reside at the junctions of functional domains because introns have evolved in part to facilitate functional domain exchanges (*Kaessmann et al., 2002*; *Patthy, 1999*). These junctions may serve as candidate internal insertion sites for FP labeling. At the same time, although CRISPIE improves upon the precision at the target site, it does not prevent potential off-target cleavages of sgRNA/Cas9. However, increasingly powerful bioinformatics tools allow for the selection of targeting sites with less likely off-target effects (e.g., *Doench et al., 2014*; *Doench et al., 2016*), and this concern will be further alleviated or eliminated in the future by the continued development of high-fidelity Cas9 variants (*Chen et al., 2017*; *Kleinstiver et al., 2016*; *Slaymaker et al., 2016*). Similarly, although CRISPIE enables the tagging of endogenous proteins with low error rates, it does not ensure that the tagged protein

functions the same as the wild-type protein. Not all tagging is benign, and rigorous characterization will be needed for each tagging experiment. It should also be noted that, when designing the donor template, care should be given to not create unintended splicing acceptor sites in the inverted orientation. Otherwise, inverted insertion events can cause mutations at the mRNA and protein levels.

CRISPIE-based DNA insertion can be readily erased, which offers flexibility in both research studies and gene therapy in the future. Although the original editing site will be destroyed during the labeling-erasing cycle, adjacent editing sites within the intron can allow for additional rounds of labeling, thereby simulating a reversible process. Overall, CRISPIE can label endogenous proteins with high fidelity, high efficiency, and reversibility in diverse animal species. This method can also conceivably be used to introduce other functional domains to modify protein function for research and for therapeutic purposes. This method may have broad applications in both biomedical research and medicine.

# Materials and methods

**Key resources table**

| Reagent type (species) or resource | Designation | Source or reference | Identifiers | Additional information |
|---|---|---|---|---|
| Gene (*Homo sapiens*) | ACTB | NA | NCBI Gene ID 60 | |
| Gene (*Homo sapiens*) | TUBA1B | NA | NCBI Gene ID 10376 | |
| Gene (*Homo sapiens*) | TOMM20 | NA | NCBI Gene ID 9804 | |
| Gene (*Homo sapiens*) | VCL | NA | NCBI Gene ID 7414 | |
| Gene (*Homo sapiens*) | CALR | NA | NCBI Gene ID 811 | |
| Gene (*Mus musculus*) | Actb | NA | NCBI Gene ID 11461 | |
| Gene (*Mus musculus*) | Tuba1b | NA | NCBI Gene ID 22143 | |
| Gene (*Mus musculus*) | Camk2a | NA | NCBI Gene ID 12322 | |
| Gene (*Rattus norvegicus*) | Actb | NA | NCBI Gene ID 81822 | |
| Gene (*Rattus norvegicus*) | Syn1 | NA | NCBI Gene ID 24949 | |
| Strain, strain background (*Rattus norvegicus*, Sprague Dawley) | Sprague Dawley rat | Charles River | Strain Code 001; RRID:RGD_734476 | |
| Genetic reagent (*Mus musculus*) | *Cas9* mouse (*B6J.129(Cg)-Igs 2$^{tm1.1(CAG-cas9*)Mmw}$/J*) | Jax | Stock No: 028239; RRID:IMSR_JAX:028239 | |
| Cell line (*Homo sapiens*) | U2OS | ATCC | Cat# HTB-96; RRID:CVCL_0042 | |
| Cell line (*Mus musculus*) | Neuro2A | ATCC | Cat# CCL-131; RRID:CVCL_0470 | |
| Recombinant DNA reagent | px330 | Addgene | Plasmid #42230 | |
| Recombinant DNA reagent | sgRNA/Cas9 targeting *ACTB* i1 | This paper | | Progenitors: synthesized oligos; pX330; will be deposited to Addgene |

*Continued on next page*

*Continued*

| Reagent type (species) or resource | Designation | Source or reference | Identifiers | Additional information |
|---|---|---|---|---|
| Recombinant DNA reagent | sgRNA/Cas9 targeting *ACTB* i2 | This paper | | Progenitors: synthesized oligos; pX330; will be deposited to Addgene |
| Recombinant DNA reagent | sgRNA/Cas9 targeting *ACTB* i3 | This paper | | Progenitors: synthesized oligos; pX330; will be deposited to Addgene |
| Recombinant DNA reagent | sgRNA/Cas9 targeting *ACTB* i4 | This paper | | Progenitors: synthesized oligos; pX330; will be deposited to Addgene |
| Recombinant DNA reagent | sgRNA/Cas9 targeting *ACTB* i5 | This paper | | Progenitors: synthesized oligos; pX330; will be deposited to Addgene |
| Recombinant DNA reagent | sgRNA/Cas9 targeting *ACTB* e1 | This paper | | Progenitors: synthesized oligos; pX330; will be deposited to Addgene |
| Recombinant DNA reagent | sgRNA/Cas9 targeting *ACTB* e2 | This paper | | Progenitors: synthesized oligos; pX330; will be deposited to Addgene |
| Recombinant DNA reagent | sgRNA/Cas9 targeting *TUBA1B* | This paper | | Progenitors: synthesized oligos; pX330; will be deposited to Addgene |
| Recombinant DNA reagent | sgRNA/Cas9 targeting *TOMM20* | This paper | | Progenitors: synthesized oligos; pX330; will be deposited to Addgene |
| Recombinant DNA reagent | sgRNA/Cas9 targeting *VCL* | This paper | | Progenitors: synthesized oligos; pX330; will be deposited to Addgene |
| Recombinant DNA reagent | sgRNA/Cas9 targeting *CALR* | This paper | | Progenitors: synthesized oligos; pX330; will be deposited to Addgene |
| Recombinant DNA reagent | sgRNA/Cas9 targeting *Actb* | This paper | | Progenitors: synthesized oligos; pX330; will be deposited to Addgene |
| Recombinant DNA reagent | sgRNA/Cas9 targeting *Tuba1b* | This paper | | Progenitors: synthesized oligos; pX330; will be deposited to Addgene |
| Recombinant DNA reagent | sgRNA/Cas9 targeting *Camk2a* | This paper | | Progenitors: synthesized oligos; pX330; will be deposited to Addgene |
| Recombinant DNA reagent | sgRNA/Cas9 targeting *rActb* | This paper | | Progenitors: synthesized oligos; pX330; will be deposited to Addgene |
| Recombinant DNA reagent | sgRNA/Cas9 targeting *rSyn1* | This paper | | Progenitors: synthesized oligos; pX330; will be deposited to Addgene |
| Recombinant DNA reagent | pUC57-Amp | Genewiz | Genewiz ID: pUC57-Amp | |
| Recombinant DNA reagent | Insertion template B1 (*Figure 1—figure supplement 3*) | This paper | | Progenitors: synthesized DNA; pUC57-Amp; will be deposited to Addgene |
| Recombinant DNA reagent | Insertion template B2 (*Figure 1—figure supplement 3*) | This paper | | Progenitors: synthesized DNA; pUC57-Amp; will be deposited to Addgene |
| Recombinant DNA reagent | Insertion template B3 (*Figure 1—figure supplement 3*) | This paper | | Progenitors: synthesized DNA; pUC57-Amp; will be deposited to Addgene |

*Continued*

| Reagent type (species) or resource | Designation | Source or reference | Identifiers | Additional information |
|---|---|---|---|---|
| Recombinant DNA reagent | Insertion template B4 (*Figure 1—figure supplement 3*) | This paper | | Progenitors: synthesized DNA; pUC57-Amp; will be deposited to Addgene |
| Recombinant DNA reagent | Insertion template B5 (*Figure 1—figure supplement 3*) | This paper | | Progenitors: synthesized DNA; pUC57-Amp; will be deposited to Addgene |
| Recombinant DNA reagent | Insertion template B6 (*Figure 1—figure supplement 3*) | This paper | | Progenitors: synthesized DNA; pUC57-Amp; will be deposited to Addgene |
| Recombinant DNA reagent | Insertion template B7 (*Figure 1—figure supplement 3*) | This paper | | Progenitors: synthesized DNA; pUC57-Amp; will be deposited to Addgene |
| Recombinant DNA reagent | Insertion template B8 (*Figure 1—figure supplement 3*) | This paper | | Progenitors: synthesized DNA; pUC57-Amp; will be deposited to Addgene |
| Recombinant DNA reagent | Insertion template B9 (*Figure 1—figure supplement 3*) | This paper | | Progenitors: synthesized DNA; pUC57-Amp; will be deposited to Addgene |
| Recombinant DNA reagent | Insertion template B10 (*Figure 1—figure supplement 3*) | This paper | | Progenitors: synthesized DNA; pUC57-Amp; will be deposited to Addgene |
| Recombinant DNA reagent | pAAV-CW3SL-EGFP | Addgene | Plasmid # 61463 | |
| Recombinant DNA reagent | pKanCMV-mRuby3-18aa-actin | Addgene | Plasmid # 74255 | |
| Recombinant DNA reagent | pKanCMV-mRuby3 | This paper | | Progenitors: oligos for site-directed deletion; pKanCMV-mRuby3-18aa-actin; will be deposited to Addgene |
| Recombinant DNA reagent | Plasmid for β-actin CRISPIE (*Figure 1—figure supplement 3C*) | This paper | | Progenitors: synthesized DNA; pKanCMV-mRuby3; pAAV-CW3SL-EGFP; will be deposited to Addgene |
| Sequence-based reagent | Oligonucleotides used for PCR and RT-PCR analysis | Genewiz | | Please see *Supplementary file 1* |
| Commercial assay or kit | NGS sequencing of PCR amplicons | Genewiz | Genewiz ID: Amplicon-EZ | |
| Chemical compound, drug | Lipofectamine 2000 | Thermo Fisher | Cat# 11668019 | |
| Chemical compound, drug | McCoy's 5A Medium (for U2OS cell culture) | ATCC | Cat# 30–2007 | |
| Chemical compound, drug | MEM (for Neuro2A cell culture) | Thermo Fisher | Cat# 11095–080 | |
| Chemical compound, drug | MEM powder (for slice culture) | Thermo Fisher | Cat# 11700–077 | |
| Chemical compound, drug | Cell culture insert (for slice cultures) | Millipore | Cat# PICM0RG50 | |

*Continued on next page*

*Continued*

| Reagent type (species) or resource | Designation | Source or reference | Identifiers | Additional information |
|---|---|---|---|---|
| Chemical compound, drug | GoScript Reverse Transcriptase | Promega | Cat# A5001 | |
| Chemical compound, drug | OneTaq Hot Start 2X Master Mix with GC Buffer | NEB | Cat# M0485S | |
| Chemical compound, drug | Q5 Hot Start High-Fidelity 2X Master Mix | NEB | Cat# M0494S | |
| Chemical compound, drug | Gold particles, 1.6 µm (for gene gun bullet) | Bio-Rad | Cat# 165–2264 | |
| Chemical compound, drug | Tefzel tubing | Bio-Rad | Cat# 165–2441 | |
| Software, algorithm | MATLAB | MathWorks | RRID:SCR_001622; https://www.mathworks.com/ | |
| Software, algorithm | GPP sgRNA designer | BROAD Institute | https://portals.broadinstitute.org/gpp/public/analysis-tools/sgrna-design | |
| Other | β-Actin CRISPIE AAV9 (*Figure 1—figure supplement 3C*) | Vigene | | Custom production |

## Plasmid constructs

Plasmid constructs were generated using gene synthesis or standard subcloning methods. All previously unpublished constructs and their corresponding sequences will be deposited to Addgene .

For most experiments, sgRNAs were constructed by cloning the targeted sequences into the pX330 vector (Addgene #42230) via the BbsI sites. sgRNAs were designed using the GPP sgRNA designer from the BROAD Institute (https://portals.broadinstitute.org/gpp/public/analysis-tools/sgrna-design). The expression of each sgRNA was driven by a U6 promotor. The same plasmid also expressed a human codon-optimized SpCas9 protein. The insertion donors were constructed based on the pUC57-Amp vector (Genewiz). Depending on the specific downstream experiments, the donors may be flanked by the same gene-targeting cutting site and/or by DRS-1 or DRS-2 sites (see *Gao et al., 2019* for the latter two; and see *Figure 1—figure supplement 3* for specific donor designs), which allows the donor to be excised by SpCas9 guided by the gene-targeting sgRNA or by an additional DRS-1 or DRS-2 sgRNA. For some templates, a U6 promoter-driven DRS-2 sgRNA was integrated into the same plasmid of the donor to ensure co-expression with the donor and efficient excision during transfection. mRuby3 (*Bajar et al., 2016*) was expressed from a pKanCMV or a pCAGGS vector.

## Cell culture and transfection

Human U2OS cells were ordered from ATCC (#HTB-96) and grown in McCoy's 5A medium (ATCC #30–2007) supplemented with 10% fetal bovine serum. Mouse Neuro 2A cells were originally from ATCC (#CCL-131) and were obtained as a gift from Dr Kevin Wright's laboratory. Note that all cells from ATCC have been authenticated by morphology, karyotyping, and PCR-based approaches. These include an assay to detect species-specific variants of the cytochrome C oxidase I gene (COI analysis) to rule out inter-species contamination and short tandem repeat profiling to distinguish between individual human cell lines and rule out intra-species contamination. These cells are also tested for mycoplasma by ATCC. Cell aliquots were kept frozen in liquid nitrogen until used and were further authenticated based on their morphology. Once thawed, each aliquot of cells was

passed and used for no more than 3 months. Both cells were grown in a tissue culture incubator at 37°C and 5% $CO_2$. Cell culture transfections were carried out using Lipofectamine 2000 (Thermo Fisher) following the standard protocol, except that only half of the recommended Lipofectamine amount (~5 μl/35 mm dish) was used with a lipofectamine to DNA ratio of 2–3:1. For confocal and SIM imaging, the cells were re-plated onto a cover-glass bottom dish (FluoroDish, WPI, #FD35-100) at 2–4 days post-transfection at an appropriate density and imaged after another 2–3 days. For *Figure 4C*, Neuro 2A cells were serum-starved (in 0.1% FBS) and treated with all-trans retinoic acid (12.5 μM) at 3 days post-transfection to induce cell differentiation and then imaged after an additional 2–3 days.

## FACS and cell cloning

Transfected U2OS cells were subjected to fluorescence-activated cell sorting (FACS) at the OHSU flow cytometry core using an Influx sorter (BD). Red label-only (i.e., transfected but not successfully labeled) and green-labeled cells were collected separately. For cell cloning, one to eight green-labeled cells were sorted into individual wells of a 96-well plate containing U2OS culture medium. Cell growth was visually inspected daily. Wells with only one to three dividing cells were monitored and later expanded into larger wells. A single-cell clone was used for *Figure 6*.

## PCR and NGS

FACS cells (30–100 k per reaction) were used. For genomic DNA preparation, cells were lysed in 0.05M NaOH and 0.1 mM EDTA at 95°C for 15 min and then neutralized using 0.1 M (final concentration) Tris-HCl. RNA was isolated following an RNeasy Micro (Qiagen) protocol utilizing a QIAcube isolation robot at the OHSU Gene Profiling Shared Resource Core. mRNA was then reverse-transcribed using the GoScript reverse transcriptase enzyme (Promega) and random primers. Nested PCR was then carried out using OneTag DNA polymerase with GC buffer and Q5 polymerase (NEB) for genomic DNA and cDNA, respectively. GC buffer was necessary for genomic DNA amplification because the relevant intron contains highly GC-rich sequences. See *Figure 2—figure supplement 1* and *Supplementary files 1* and *2* for primer location, sequences, and combinations. A 4 bp barcode was added to the end of second-round primers to distinguish between individual samples. The first round of PCR for different products of each sample was carried out in a single reaction by including all relevant primers and was amplified for 23 cycles with an annealing temperature of 60°C. Approximately 2 μl of 10-fold diluted PCR products (to minimize the carryover of first-round primers) was added to 50 μl reactions for second-round PCRs with nested, specific primer pairs for 23 cycles. The exact added amount of first-round PCR product was adjusted so that different samples gave comparable band intensities for the control PCR for *TUBA1B*. The PCR amplicons from different samples were mixed and submitted for NGS (Genewiz). The sequencing results were unmixed by the barcodes and by the presence of specific primers. INDELs were identified using custom code written in MATLAB.

## Hippocampal slice culture and transfection

Hippocampi were dissected from P6–P7 Sprague Dawley rat pups (both sexes) and sectioned to 400 μm slices in dissection medium containing (in mM) 1 $CaCl_2$, 5 $MgCl_2$, 10 D-glucose, 4 KCl, 26 $NaHCO_3$, and 248 sucrose, with addition of 0.00025% phenol red. The slices were then seeded to a cell culture insert (Millipore, #PICM0RG50) and cultured at 35°C with 5% $CO_2$ in medium containing 7.4 g/l MEM (Thermo Fisher #11700–077) with the following supplements (in mM unless labeled otherwise): 16.2 NaCl, 2.5 L-Glutamax, 0.58 $CaCl_2$, 2 $MgSO_4$, 12.9 D-glucose, 5.2 $NaHCO_3$, 30 HEPES, 0.075% ascorbic acid, 1 μg/ml insulin, and 20% heat-inactivated horse serum. Organotypic hippocampal slice cultures were transfected using the biolistic method. Plasmids were coated onto 1.6 μm gold beads, and the slices were transfected at 4–7 days in vitro using a Helios gene gun (Bio-Rad). Slices were examined at 1–2 weeks post-transfection.

## Animals, in vivo transfections, and sample preparations

Animal handling and experimental protocols were performed in accordance with the recommendations in the Guide for the Care and Use of Laboratory Animals of the National Institutes of Health and were approved by the Institutional Animal Care and Use Committee (IACUC) of the Oregon

Health and Science University (#IS00002792). *SpCas9* homozygous mice were crossed with C57BL/6 mice (Charles River). The resulting heterozygous offspring of both sexes were transfected using either IUE (at E16) or by stereotaxic viral injection (p14–60). In both cases, the somatosensory cortex was targeted. For in vivo imaging in living mice, a glass window was installed on the skull of the mouse at 1–4 months of age, as described previously (*Ma et al., 2018*). In vivo imaging was performed immediately after the window was installed.

IUE was performed at E16 as described previously (*Ma et al., 2018*) by injecting plasmid DNA (1 µl/embryo, total DNA concentration ~3–4 µg/µl, mixed with a 0.2% final concentration of fast green for visualization) into the lateral ventricle of mouse embryos, which were then electroporated with five 100 ms pulses (38 V) using an electroporator (BEX #CUY21). Stereotaxic viral injections were performed as previously described (*Hunnicutt et al., 2014*) at ~p21 using a custom-modified David Kopf system. Typically, 20–50 nl of AAV was injected.

## Acute brain slices

Mice (p18–50) with neurons transfected using IUE were cardiac perfused with ice-cold, gassed (with 95% $O_2$/5% $CO_2$) artificial cerebral spinal fluid (ACSF) containing (in mM) 127 NaCl, 25 NaHCO$_3$, 25 D-glucose, 2.5 KCl, 1.25 NaH$_2$PO$_4$, 2 CaCl$_2$, and 1 MgCl$_2$. The brain was then dissected and coronal slices were prepared using a vibratome (Leica VT1200S) in a choline cutting solution (gassed with 95% $O_2$/5% $CO_2$) containing (in mM) 110 choline chloride, 25 NaHCO$_3$, 25 D-glucose, 2.5 KCl, 7 MgCl$_2$, 0.5 CaCl$_2$, 1.25 NaH$_2$PO$_4$, 11.5 sodium ascorbate, and 3 sodium pyruvate. The slices were the incubated in gassed ACSF at 35°C for 30 min and subsequently kept at room temperature for up to 6 hr.

## Imaging and image analysis

All imaging experiments were carried out on live samples. Cultured cells were imaged using a custom-built two-photon microscope (up-right) (*Ma et al., 2018*), a Zeiss LSM 880 Airyscan laser scanning confocal microscope, or an Elyra 7 lattice-based structured illumination microscope. Cultured and acute brain slices were imaged in a chamber perfused with gassed ACSF. In vivo imaging was performed using a custom-built two-photon microscope based on the Janelia MIMMS design through a cranial window over transfected neurons in the mouse cortex. For two-photon microscopy, samples were excited using a Maitai HP Ti:Sapphire laser (Newport) at 960 or 990 nm, and green and red fluorescence were unmixed using a dichroic (Chroma 565DCXR) and band-pass filters (Chroma ET500/40 for green and Semrock FF01-630/92 for red). For Airyscan confocal microscopy and SIM, setups were configured according to the manufacturer's suggestions for Nyquist spatial sampling, and raw data was auto-processed. Throughout the manuscript, red fluorescence is presented as magenta. For the cell growth assay, transfected cells were re-plated to coverslip-bottomed six-well plates, and were maintained and imaged using a Zeiss Celldiscoverer imaging system.

Image analyses were performed using custom software written in MATLAB. Labeling efficiencies were calculated as successful CRISPIEd cells (green) divided by total transfected cells (red) in random field of views (FOVs) (2 p imaging; 330 × 330 or 165 × 165 µm$^2$ FOV) selected based on the red channel only. In some experiments, the numbers of total CRISPIEd (green) and transfected (red) cells were visually counted from nine FOVs of predetermined arrangement (a 3 × 3 grid separated by 600 µm) under a 40× objective. All conditions to be compared were transfected and imaged side-by-side, and data were normalized to the averaged cell counts per FOV of the best labeled transfections. For IUE experiments, only the centers of the transfected regions were imaged and quantified. For mTurquoise2 and mNeonGreen labeling, the images were taken under the same conditions as mEGFP but were pseudo-colored in the color of turquoise and neon green, respectively. For the cell growth assay, labeled cells were manually counted. To control for potential cell damage/death during passaging of the cells before imaging, only CRISPIE-labeled cells that divided at least once were included for analyses. Untransfected cells within a region of interest (ROI) (~1 × 0.8 mm; 13–65 cells at time zero) with the same FOV as labeled cells were also counted.

SEI measurements were calculated as described previously (*Zhong et al., 2009*). Briefly, averaged green and red intensities were measured from manually drawn ROIs on the spines and their adjacent parental dendritic shaft in 2 p images. SEI was then calculated as $\log_2[(F_{green}/F_{red})_{spine}/(F_{green}/$

$F_{red})_{shaft}$], where F is the average fluorescence intensities of individual ROIs. Positive values indicate relative spine enrichment, whereas negative values indicate relative spine exclusion.

## Patch-clamp electrophysiology

Whole-cell patch-clamp was performed in CRISPIEd CA3 pyramidal neurons and adjacent untransfected neurons (within 50 μm) in organotypic cultured slices. Voltage-clamp recordings were performed using a MultiClamp 700B amplifier (Molecular Devices) controlled with custom software written in MATLAB. Electrophysiological signals were acquired/digitized at 20 kHz and filtered at 2 kHz. Slices were perfused at room temperature with ACSF containing (mM): NaCl (127), $NaHCO_3$ (25), $NaH_2PO_4$ (1.25), glucose (25), KCl (2.5), $CaCl_2$ (4), $MgCl_2$ (4), pH ~7.3, and 310 mOsmol/kg when gassed with carbogenic mixture (95% $O_2$, 5% $CO_2$). Recording pipettes (3–5 MΩ) were pulled from borosilicate glass (G150F-3; Warner Instruments) using a model P-1000 puller (Sutter Instruments). Series resistance was 10–25 MΩ. The internal solution contained (in mM): Cs-gluconate (126), CsCl (10), HEPES (10), Na-phosphocreatine (5), Na-GTP (0.5), Mg-ATP (4), TEA-Cl (10), EGTA (5), and QX-314 bromide (4) with an osmolarity of 295–300 mOsmol/kg and pH ~7.2 adjusted with KOH. The liquid junction potential (*ca.* −14 mV calculated using JCal from the Clampex software, Molecular Devices) was not corrected. mEPSCs were measured at −70 mV in the presence of 1 μM TTX, 10 μM CPP, 100 μM picrotoxin. Ten-minute traces were recorded and mEPSCs events were detected using the template matching feature of Clampfit (Molecular Devices).

## Sample size, replication, and sample allocation

Sample sizes, as indicated in figure legends, were determined based on the observed variability across measurements. All experiments have been replicated in multiple independent trials/animals (typically >3) with the exception that only one round of NGS was carried out. Cells and animals were randomly allocated to each experimental groups. All groups to be compared were carried out side-by-side. The researchers were not blinded to group allocation during data collection and analysis. However, the FOVs were selected either predetermined or in the absence of knowledge of the color channel under investigation.

## Quantification, presentation, and statistics

Quantification and statistical tests were performed using MATLAB scripts. p-Values were obtained from one-way ANOVA tests, unless noted otherwise. In all figures, *$p \leq 0.05$ and is statistically significant after Bonferroni correction for multiple tests, **$p \leq 0.01$, and ***$p \leq 0.001$.

# Acknowledgements

We thank Dr Zefeng Wang for discussions regarding exon-intron junctions; Drs Paul Meraner and Marc Freeman for the *SpCas9* mouse line; Dr Kevin Wright for the Neuro 2A cell line; Dr Bart Jongbloets for help with imaging, Dr James Frank for help with experimental attempts that are not included, Mr Matthew Schleisman at the OHSU flow cytometry core, and Ms Brittany Daughtry at the OHSU Gene Profiling Shared Resource for technical assistance on FACS and mRNA purification experiments, respectively. We thank Dr Gail Mandel for critical comments on the manuscript. The work was supported by an NIH/BRAIN Initiative grant (RF1MH120119) to HZ and TM, and an NIH/NINDS R01 (R01NS081071) to TM.

# Additional information

### Competing interests

Tianyi Mao: Reviewing editor, *eLife*. The other authors declare that no competing interests exist.

### Funding

| Funder | Grant reference number | Author |
| --- | --- | --- |
| NIH | RF1MH120119 | Haining Zhong<br>Tianyi Mao |

| NINDS | R01NS081071 | Tianyi Mao |

The funders had no role in study design, data collection and interpretation, or the decision to submit the work for publication.

## Author contributions
Haining Zhong, Conceptualization, Data curation, Software, Formal analysis, Supervision, Funding acquisition, Validation, Investigation, Visualization, Methodology, Writing - original draft, Project administration, Writing - review and editing; Cesar C Ceballos, Maozhen Qin, Investigation; Crystian I Massengill, Michael A Muniak, Lei Ma, Investigation, Writing - review and editing; Stefanie Kaech Petrie, Data curation, Investigation, Visualization, Writing - review and editing; Tianyi Mao, Conceptualization, Supervision, Funding acquisition, Writing - original draft, Writing - review and editing

## Author ORCIDs
Haining Zhong (iD) https://orcid.org/0000-0002-7109-4724
Michael A Muniak (iD) http://orcid.org/0000-0001-8047-5871
Tianyi Mao (iD) http://orcid.org/0000-0002-3532-8319

## Ethics
Animal experimentation: Animal handling and experimental protocols were performed in accordance with the recommendations in the Guide for the Care and Use of Laboratory Animals of the National Institutes of Health and were approved by the Institutional Animal Care and Use Committee (IACUC) of the Oregon Health & Science University (#IS00002792).

## Decision letter and Author response
Decision letter https://doi.org/10.7554/eLife.64911.sa1
Author response https://doi.org/10.7554/eLife.64911.sa2

# Additional files

## Supplementary files
- Supplementary file 1. PCR primer list.
- Supplementary file 2. PCR primer combinations.
- Supplementary file 3. CRISPIE protocol.
- Transparent reporting form

## Data availability
All data are included in the manuscript and supporting source data files. Source data files have been provided for Figures 1, 2, 3, 5, and 6.

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
