## [Decision Letter]

**Acceptance summary:**

The design of the genomic insertion into introns tolerates INDELs and increases accurate targeting in somatic cells. The authors provided evidence that several in situ fusion proteins remain functional.

**Decision letter after peer review:**

Thank you for submitting your article "High-fidelity, efficient, and reversible labeling of endogenous proteins using CRISPR-based designer exon insertion" for consideration by *eLife*. Your article has been reviewed by 2 peer reviewers, and the evaluation has been overseen by a Reviewing Editor and Richard Aldrich as the Senior Editor. The following individual involved in review of your submission has agreed to reveal their identity: Craig C Mello (Reviewer #1).

Essential revisions:

1. What is new is the finding that FP insertions are frequently expressed and at least partly functional as evidenced by their ability to localize to the expected intracellular structures. However, no actual functional data is provided in this study so it remains to be seen how frequently the insertion of FP exons is tolerated. It would help the study substantially to have functional information for a few insertions.

The value and utility of this study hinges on whether insertions of this type frequently retain function. The authors speculate that "labeling at an internal site of a gene is feasible as long as the insertion does not disrupt the function of the encoded protein. Many introns reside at the junctions of functional domains because introns have evolved in part to facilitate functional domain exchanges (Kaessmann et al., 2002; Patthy, 1999)." Thus an analysis of how often intron tags are tolerated as homozygotes would be helpful for users who will worry that a potentially "quick and dirty" CRISPIE insertion might not accurately report on the function and localization of their protein of interest.

2. Adoption of this approach by the community will depend on access to reagents and protocols. I applaud the authors for stating that they plan to deposit plasmids at Addgene (lines 349-350). However, they also need to provide a list or table of plasmids that will be deposited, along with a description of the purpose of each plasmid, to allow potential adopters to quickly figure out which constructs are needed. From the schematic provided in Figure 1—figure supplement 3, it appears that at least some of the constructs (b1-b5) must be modified prior to use to insert an sgRNA target that will release the synthetic exon from its plasmid; others (b6-b10) might not require any modification before transfection, but this needs to be clearly stated and explained. Do b6-b10 represent newer versions of the design that can be used without any cloning required? Which construct should a novice user choose when trying this approach for the first time? To answer these and related questions, I would urge the authors to provide a detailed protocol for targeting a new gene of interest – including all cloning steps – as a supplemental text file.

3. The results in figure 2B show that a population of CRISPIEd cells contains alleles with no insertion, correct insertions, and inverted insertions. I really wanted to know the relative abundance of these – especially the frequency of inverted insertions compared to correct, forwards insertions. This could be easily measured using qPCR of genomic DNA, and should be reported for several different targeted loci (and ideally in several different cell lines) to get a sense of how variable the insertions are. This is critical information to enable a reader to evaluate whether the CRISPIE approach will be useful for a particular application.

4. The high incidence of indel mutations at the 5' end of forward inserts (Figure 2D-iv, top row) is surprising, since previous reports have suggested that indels are relatively uncommon when using the NHEJ approach to target exons (Suzuki et al. Nature 2016; Artegiani et al. Nature Cell Biol. 2020). The authors don't provide an explanation for this difference – could it be a specific consequence of targeting introns? Or is it rather due to experimental differences such as choice of cell line, transfected plasmid concentration, etc.? Did the authors observe a similar indel frequency when targeting exons (in the experiment reported in Figure 3)?

5. Please report the data in Figures 3B-C as absolute labeling efficiencies instead of relative efficiencies. Normalizing efficiency to 1 is misleading and obscures the actual frequency of successful labeling.

*Reviewer #1 (Recommendations for the authors):*

The gene therapy literature has been touting introns as ideal targets for the very same reasons that you are here. So you should acknowledge them.

In line 144 you say "Had the coding sequence been targeted, an inverted label insertion would inevitably cause disruptive mutations. However, under our conditions, only wild-type and forwardly inserted mRNAs were detected (Figure 2B), demonstrating the advantage of CRISPIE." Rather than saying "our conditions" for clarity you should explain that the donor only contains splicing signals for new exon inclusion in one orientation. And so the mRNA is altered only by the forward insertion.

You mention that "to the best of our knowledge, this is the first demonstration of a readily reversible gene editing approach at the DNA level." The concept of reversibility is very well established in genome editing going way back to using loxP sites and Cre drivers. Numerous other CRISPR studies have described the incorporation of new sgRNA target sites for just this reason also. What you mean to say is that it is easier than these other CRISPR methods since you are in an intron where indels from DSBs will be better tolerated.

*Reviewer #2 (Recommendations for the authors):*

The authors should address the following points in their revision:

– Adoption of this approach by the community will depend on access to reagents and protocols. I applaud the authors for stating that they plan to deposit plasmids at Addgene (lines 349-350). However, they also need to provide a list or table of plasmids that will be deposited, along with a description of the purpose of each plasmid, to allow potential adopters to quickly figure out which constructs are needed. From the schematic provided in Figure 1—figure supplement 3, it appears that at least some of the constructs (b1-b5) must be modified prior to use to insert an sgRNA target that will release the synthetic exon from its plasmid; others (b6-b10) might not require any modification before transfection, but this needs to be clearly stated and explained. Do b6-b10 represent newer versions of the design that can be used without any cloning required? Which construct should a novice user choose when trying this approach for the first time? To answer these and related questions, I would urge the authors to provide a detailed protocol for targeting a new gene of interest – including all cloning steps – as a supplemental text file.

– The results in figure 2B show that a population of CRISPIEd cells contains alleles with no insertion, correct insertions, and inverted insertions. I really wanted to know the relative abundance of these – especially the frequency of inverted insertions compared to correct, forwards insertions. This could be easily measured using qPCR of genomic DNA, and should be reported for several different targeted loci (and ideally in several different cell lines) to get a sense of how variable the insertions are. This is critical information to enable a reader to evaluate whether the CRISPIE approach will be useful for a particular application.

– I was surprised by the high incidence of indel mutations at the 5' end of forward inserts (Figure 2D-iv, top row), since previous reports have suggested that indels are relatively uncommon when using the NHEJ approach to target exons (Suzuki et al. Nature 2016; Artegiani et al. Nature Cell Biol. 2020). The authors don't provide an explanation for this difference – could it be a specific consequence of targeting introns? Or is it rather due to experimental differences such as choice of cell line, transfected plasmid concentration, etc.? Did the authors observe a similar indel frequency when targeting exons (in the experiment reported in Figure 3)?

– Please report the data in Figures 3B-C as absolute labeling efficiencies instead of relative efficiencies. Normalizing efficiency to 1 is misleading and obscures the actual frequency of successful labeling.

---

## [Author Response]

Essential revisions:1. What is new is the finding that FP insertions are frequently expressed and at least partly functional as evidenced by their ability to localize to the expected intracellular structures. However, no actual functional data is provided in this study so it remains to be seen how frequently the insertion of FP exons is tolerated. It would help the study substantilly to have functional information for a few insertions.The value and utility of this study hinges on whether insertions of this type frequently retain function. The authors speculate that "labeling at an internal site of a gene is feasible as long as the insertion does not disrupt the function of the encoded protein. Many introns reside at the junctions of functional domains because introns have evolved in part to facilitate functional domain exchanges (Kaessmann et al., 2002; Patthy, 1999)." Thus an analysis of how often intron tags are tolerated as homozygotes would be helpful for users who will worry that a potentially "quick and dirty" CRISPIE insertion might not accurately report on the function and localization of their protein of interest.

We now provide the results of three different experiments to examine the function of CRISPIEd β-actin, and, in one of the experiments, CRISPIEd α-tubulin 1B. The functions of the cytoskeleton is intimated linked to cell growth. We now show that CRISPIE labeling of β-actin, using two different intronic loci, and CRISPIE labeling of α-tubulin 1B (*TUBA1B*) do not affect the growth of U2OS cells (new Experiment #1; Figure 1H, and Figure 1—figure supplement 4), suggesting that the functions of these two FP-labeled cytoskeletal proteins are unperturbed. In addition, as suggested, we now demonstrate that cells that are homozygous for the CRISPIE insertion are viable and able to divide (new Experiment #2; Figure 4—figure supplement 1). We also show that two important neuronal functional parameters – the mEPSC frequency and amplitude – are not altered (new Experiment #3; Figure 5—figure supplement 2).

We also hope to emphasize that, although CRISPIE provides a way to perform FP tagging of endogenous proteins with high efficiency and low error rates, it cannot ensure that the FP-tagging itself is benign for all proteins. Numerous studies have overexpressed FP-tagged proteins, which is well-documented to be associated with potential side effects. The CRISPIE method empowers researchers by allowing them to tag endogenous proteins without overexpression. However, if the FP-tagging itself affects protein function, CRISPIE will not be helpful. Each FP-tagging project, whether it is based on CRISPIE or any other methods, will requires its own systematic characterization. We have now made this clear in the discussion (pg. 17): “… although CRISPIE enables the tagging of endogenous proteins with low error rates, it does not ensure that the tagged protein functions the same as the wild-type protein. Not all tagging is benign, and rigorous characterizations will be needed for each tagging experiment.”

2. Adoption of this approach by the community will depend on access to reagents and protocols. I applaud the authors for stating that they plan to deposit plasmids at Addgene (lines 349-350). However, they also need to provide a list or table of plasmids that will be deposited, along with a description of the purpose of each plasmid, to allow potential adopters to quickly figure out which constructs are needed. From the schematic provided in Figure 1—figure supplement 3, it appears that at least some of the constructs (b1-b5) must be modified prior to use to insert an sgRNA target that will release the synthetic exon from its plasmid; others (b6-b10) might not require any modification before transfection, but this needs to be clearly stated and explained. Do b6-b10 represent newer versions of the design that can be used without any cloning required? Which construct should a novice user choose when trying this approach for the first time? To answer these and related questions, I would urge the authors to provide a detailed protocol for targeting a new gene of interest – including all cloning steps – as a supplemental text file.

We thank the reviewers and editors for this suggestion. We are committed to help the community to use our method. In fact, we have already received and responded to many inquiries throughout the world after the manuscript was deposited in BioRxiv. All plasmids that have not been previously reported will be deposited to Addgene. The list to be deposited is now included in the Key Resource Table at the beginning of Materials and methods. We have now also provided a detailed step-by-step protocol (new documentation). We believe that the protocol should give researchers a realistic grasp of how the CRISPIE method is carried out. We will be happy to modify or amend the protocol further should the reviewers have additional suggestions.

The constructs described in B6-B10 (Figure 1—figure supplement 3) were from an effort to make the technique more standardized. They can be used for initial testing of conditions. However, empirically they give lower efficiency than target-specific donors (e.g., those described in B1-B5), presumably due to the loss of preferential insertion into the edited locus. We suggest that users construct target-specific donors when repeated experiments or higher labeling efficiencies are needed. We have now described this in the new detailed protocol. We write (in the protocol): “For initial testing, identify a generic donor vector of the appropriate reading frame (phase) that expresses the appropriate FP. […] We empirically find that target-specific donors give a 50 – 100% higher labeling efficiency compared to generic donors.”

3. The results in figure 2B show that a population of CRISPIEd cells contains alleles with no insertion, correct insertions, and inverted insertions. I really wanted to know the relative abundance of these – especially the frequency of inverted insertions compared to correct, forwards insertions. This could be easily measured using qPCR of genomic DNA, and should be reported for several different targeted loci (and ideally in several different cell lines) to get a sense of how variable the insertions are. This is critical information to enable a reader to evaluate whether the CRISPIE approach will be useful for a particular application.

As described in detail in the next paragraph, based on our preliminary results, we found that, although qPCR can determine the relative abundance of the same DNA species across samples, it is difficult to use qPCR to compare the relative abundances of different DNA species within a sample. Adding to the difficulty, CRISPIE targets introns, where the DNA is often high in GC content or has secondary structures. All of these issues combine to make the requested experiment extremely challenging (see details below). We hope that the reviewers see that, regardless of the results from this experiment, they will not change the conclusion that CRISPIE allows for FP labeling of endogenous proteins with much reduced error rates at both the mRNA and protein levels. As we have previously demonstrated, the inverted insertions, although they occur at the genomic DNA level, will be spliced out and will not be transcribed into mRNAs (Figure 2B).

Practically, different DNA sequences and different primers can exhibit different PCR amplification rates. This means that, after each PCR cycle, the PCR product does not always precisely double. Because PCR amplification of DNA is exponential, even small differences in amplification rates can accumulate to give significant differences after 30 to 40 cycles. For example, if there are two hypothetical PCR reactions, including one reaction in which the product doubles every cycle, and one reaction where the product increases by 95% every cycle (i.e., 5% less amplification per cycle), the differences in amplification become greater than 2-fold after 30 cycles. Adding to the difficulty is that intronic DNA sequences often contain highly GC-rich sequences and/or secondary structures, making them very difficult to amplify faithfully and fully. However, the nature of the requested experiment does not allow us to select other DNA regions of the same gene to avoid difficult regions (i.e., PCR amplification has to be at the edited locus).

Specifically, in our preliminary testing, we used online tools (IDTDNA) to identify optimal primers and DNA sizes for qPCR. However, for *hACTB* labeling (our model for demonstrating the CRISPIE method), the targeted region is ~80% GC-rich and cannot be specifically amplified in 40 cycles, regardless of whether the locus is edited or not. A control region located at a later, non-GC-rich region of *hACTB* can be amplified in less than 30 cycles under the same PCR conditions. In fact, for Figure 2, we had to use nested PCR to ensure the specificity, and amplify using 46 cycles with a special “GC buffer”, which is not available for current qPCR reagents (Applied Biosystems). We also tested two PCR sites of another gene, *hTUBA1B*. In wild-type unedited samples, both sites can be amplified by qPCR. However, their ΔCt is ~ 1.5 cycles different from each other. Since this is an unedited sample, the two sites of the same gene must have the same abundance in the starting material. This observed difference in PCR amplification again points out that different loci/primers will be amplified differently using PCR. It is therefore difficult to compare the relative abundances of different DNA insertions, as they will have different GC content and different primers will be used.

4. The high incidence of indel mutations at the 5' end of forward inserts (Figure 2D-iv, top row) is surprising, since previous reports have suggested that indels are relatively uncommon when using the NHEJ approach to target exons (Suzuki et al. Nature 2016; Artegiani et al. Nature Cell Biol. 2020). The authors don't provide an explanation for this difference – could it be a specific consequence of targeting introns? Or is it rather due to experimental differences such as choice of cell line, transfected plasmid concentration, etc.? Did the authors observe a similar indel frequency when targeting exons (in the experiment reported in Figure 3)?

As suggested, we have now performed next-generation sequencing analyses for INDELs at four different insertion sites (two intronic and two exonic, at the 5’ end of the insertion) of *hACTB* in both U2OS and HEK 293 cells (new Experiment #4; Figure 2—figure supplement 2). We also do so for one site of anther gene, *hTUBA1B*, in U2OS cells. We found that the INDEL rates are high in general, although there are indeed cell- and locus-dependent differences (see also Shen et al., Nature 2018).

We do not know exactly why the levels of INDELs in Suzuki et al. and Artegiani et al. are lower than our numbers. The specific subcellular locus that is targeted may play a role (e.g., Shen et al., Nature 2018). In Artegiani et al., the unlabeled alleles (i.e., no insertion) also have few INDELs. However, CRISPR editing is known for creating INDELs when an insertion is not involved, because correct repairs at the target sequence will be cut again and again until an INDEL occurs. This result from Artegiani et al. is also in contrast with that of Suzuki et al., which reports that “the rest of the genomic targets [the ones without insertion] all contained indels”. Regardless of what the reason is, these inconsistencies suggest that the levels of measurable INDELs may be sensitive to experimental conditions. Both Suzuki et al. and Artegiani et al. grew the cells/organoids before picking successfully grown cell clones/organoids for PCR and sequencing. We wonder whether INDELs may have put certain pressures that affect cell growth. In CRISPIE, INDELs in introns are better tolerated. We also do not expand the cells prior to PCR. We perform PCR from cell mixes either immediately after cell sorting (Figure 2), or from the whole dish without sorting (Figure 2—figure supplement 2).

Despite the difference in exact numbers, Suzuki et al. also reports ~40% INDELs at either one or both ends of insertion when their “2C” template is used (Extended Figure 2g in Suzuki et al.). This “2C” template is similar to our template except that it targets exons. As quoted above, the majority of transfected but unlabeled alleles contain INDELs, which can cause mutations at the mRNA level. At these INDEL rates, CRISPIE has significant advantages over the other methods by greatly lowering the error rate at both the mRNA and protein levels (< 2% error rate across conditions; Figure 2 of our manuscript).

5. Please report the data in Figures 3B-C as absolute labeling efficiencies instead of relative efficiencies. Normalizing efficiency to 1 is misleading and obscures the actual frequency of successful labeling.

As suggested, we have redone the experiment and we now provide the absolute labeling efficiency (new Experiment #5; Figure 3B and 3C). The results are qualitatively similar to the previous results, although the exact number is slightly different (e.g., the difference between intron and exon labeling is now > 5-fold instead of the previously reported 4-fold difference).

Reviewer #1 (Recommendations for the authors):The gene therapy literature has been touting introns as ideal targets for the very same reasons that you are here. So you should acknowledge them.

We thank the reviewer for pointing out this omission. We now cite Bernarski et al., 2016, which describes the use of super-exons in gene therapy. We write (pg. 15): “intronic editing has been used to introduce non-coding DNA tags, microRNAs, gene disruptions, exon replacements, and super-exons (Bednarski et al., 2016; Chen et al., 2018; Jarvik et al., 1996; Lee et al., 2018; Li et al., 2016; Miura et al., 2015)…”.

In line 144 you say "Had the coding sequence been targeted, an inverted label insertion would inevitably cause disruptive mutations. However, under our conditions, only wild-type and forwardly inserted mRNAs were detected (Figure 2B), demonstrating the advantage of CRISPIE." Rather than saying "our conditions" for clarity you should explain that the donor only contains splicing signals for new exon inclusion in one orientation. And so the mRNA is altered only by the forward insertion.

We have now clarified in our revised manuscript, as suggested. We write (pg. 8): “However, under our conditions, because the donor only contained the splicing signals for exon inclusion in the forward orientation, only wild-type and forwardly inserted mRNAs were detected (Figure 2B), demonstrating the advantage of CRISPIE.”

You mention that "to the best of our knowledge, this is the first demonstration of a readily reversible gene editing approach at the DNA level." The concept of reversibility is very well established in genome editing going way back to using loxP sites and Cre drivers. Numerous other CRISPR studies have described the incorporation of new sgRNA target sites for just this reason also. What you mean to say is that it is easier than these other CRISPR methods since you are in an intron where indels from DSBs will be better tolerated.

We thank the reviewer for pointing this out. We have removed the “first” claim throughout the manuscript.

Reviewer #2 (Recommendations for the authors):The authors should address the following points in their revision:– Adoption of this approach by the community will depend on access to reagents and protocols. I applaud the authors for stating that they plan to deposit plasmids at Addgene (lines 349-350). However, they also need to provide a list or table of plasmids that will be deposited, along with a description of the purpose of each plasmid, to allow potential adopters to quickly figure out which constructs are needed. From the schematic provided in Figure 1—figure supplement 3, it appears that at least some of the constructs (b1-b5) must be modified prior to use to insert an sgRNA target that will release the synthetic exon from its plasmid; others (b6-b10) might not require any modification before transfection, but this needs to be clearly stated and explained. Do b6-b10 represent newer versions of the design that can be used without any cloning required? Which construct should a novice user choose when trying this approach for the first time? To answer these and related questions, I would urge the authors to provide a detailed protocol for targeting a new gene of interest – including all cloning steps – as a supplemental text file.

We thank the reviewer for this suggestion. We are committed to helping the community to use our method. In fact, we have already received and responded to many inquiries throughout the world after the manuscript was deposited in BioRxiv. All major plasmids that have not been previously reported will be deposited to Addgene. The list to be deposited is now included in the Key Resource Table. We have now also provided a detailed step-by-step protocol (new documentation). We believe that this protocol should give researchers a realistic grasp of how the CRISPIE method is carried out. We will be happy to modify or amend the protocol further should the reviewers have additional suggestions.

The constructs described in B6-B10 (Figure 1—figure supplement 3) are the result of our efforts to make the technique more standardized. They can be used for initial testing of conditions. However, empirically they give lower efficiency than target-specific donors (e.g., those described in B1-B5), presumably due to the loss of preferential insertion into the edited locus. We suggest that users construct sgRNA-specific donors when repeated experiments or higher labeling efficiencies are needed.

We have now described this in the new detailed protocol. We write (in protocol): “For initial testing, identify a generic donor vector of the appropriate reading frame (phase) that expresses the appropriate FP. […] We empirically find that target-specific donors give a 50 – 100% higher labeling efficiency compared to generic donors.”

– The results in figure 2B show that a population of CRISPIEd cells contains alleles with no insertion, correct insertions, and inverted insertions. I really wanted to know the relative abundance of these – especially the frequency of inverted insertions compared to correct, forwards insertions. This could be easily measured using qPCR of genomic DNA, and should be reported for several different targeted loci (and ideally in several different cell lines) to get a sense of how variable the insertions are. This is critical information to enable a reader to evaluate whether the CRISPIE approach will be useful for a particular application.

We have attempted the suggested experiments. However, as described in detail in the next paragraph, we find that, although qPCR can determine the relative abundance of the same DNA species across samples, it falls short when comparing the relative abundances between different DNA species within a sample. Adding to the difficulty, CRISPIE targets introns, where the DNA is often high in GC content or has secondary structures. All of these hurdles combine to make the requested experiment extremely challenging (see details below). We hope the reviewers recognize that, regardless of the results from this experiment, they will not change the conclusion that CRISPIE allows for FP labeling of endogenous proteins with much reduced error rates at the mRNA and protein levels. As we have previously demonstrated, the inverted insertions, although they occur at the genomic DNA level, will be spliced out and will not be transcribed into mRNAs (Figure 2B).

Different DNA sequences and different primers can exhibit different PCR amplification rates. In other words, after each PCR cycle, the PCR product does not always precisely double. Because PCR’s amplification is exponential, even small differences in amplification rates can accumulate to give rise to significant differences after 30 to 40 cycles. For example, if there are two hypothetical PCR reactions, including one reaction in which the product doubles every cycle, and one reaction in which the product increases by 95% per cycle (i.e., 5% less amplification per cycle), the differences in amplification become more than 2-fold after 30 cycles. Adding to the difficulty, intronic DNA sequences often contain highly GC-rich sequences and/or secondary structures, making them very difficult to amplify faithfully and fully. However, the nature of the requested experiment does not allow us to select other DNA regions of the same gene to avoid difficult regions (i.e., PCR amplification has to be at the edited locus).

Specifically, in our preliminary testing, we used online tools (IDTDNA) to identify optimal primers and DNA sizes for qPCR. However, for *hACTB* labeling (our prototype for demonstrating the CRISPIE method), the targeted region is ~80% GC-rich and cannot be specifically amplified in 40 cycles, regardless of whether the locus is edited or not. A control region located at a later, non-GC-rich part of *hACTB* can be amplified in less than 30 cycles under the same PCR conditions. In fact, for Figure 2, we had to use nested PCR with to ensure the specificity, and amplify using 46 cycles with a special “GC buffer”, which is not available for current qPCR reagents (Applied Biosystems). We also tested two PCR sites of another gene, *hTUBA1B*. In wild-type unedited samples, both sites can be amplified by qPCR. However, their ΔCt is ~ 1.5 cycles different from each other. Since this is an unedited sample, the two sites of the same gene must have the same abundance in the starting material. This observed difference in PCR amplification again points out that different loci/primers will be amplified differently using PCR. It is therefore difficult to compare the relative abundances of different DNA insertions, as they will have different GC content, and different primers will be used.

– I was surprised by the high incidence of indel mutations at the 5' end of forward inserts (Figure 2D-iv, top row), since previous reports have suggested that indels are relatively uncommon when using the NHEJ approach to target exons (Suzuki et al. Nature 2016; Artegiani et al. Nature Cell Biol. 2020). The authors don't provide an explanation for this difference – could it be a specific consequence of targeting introns? Or is it rather due to experimental differences such as choice of cell line, transfected plasmid concentration, etc.? Did the authors observe a similar indel frequency when targeting exons (in the experiment reported in Figure 3)?

As suggested, we have now performed next-generation sequencing analyses for INDELs at four different insertion sites (two intronic and two exonic, at the 5’ end of the insertion) of *hACTB* in both U2OS and HEK 293 cells (new Experiment #4; Figure 2—figure supplement 2). We also do so for one site of anther gene, *hTUBA1B*, in U2OS cells. We found that the INDEL rates are high in general, although there are indeed cell- and locus-dependent differences (see also Shen et al., Nature 2018).

We do not know exactly why the levels of INDELs in Suzuki et al. and Artegiani et al. are lower than our numbers. The specific subcellular locus may play a role (e.g., Shen et al., Nature 2018). In Artegiani et al., the unlabeled alleles (i.e., no insertion) also have few INDELs. However, CRISPR editing is known for creating INDELs when insertion is not involved, because correct repairs at the target sequence will be cut again and again until an INDEL occurs. This result of Artegiani et al. is also in contrast with that of Suzuki et al., which reports that “the rest of the genomic targets [the ones without insertion] all contained indels”. Regardless of the reason, these inconsistencies suggest that the level of measurable INDELs may be sensitive to experimental conditions. Both Suzuki et al. and Artegiani et al. grew the cells/organoids before picking successfully grown cell clones/organoids for PCR and sequencing. It is possible that INDELs may have put certain pressures that affect growth. In CRISPIE, INDELs in introns are better tolerated. We also do not expand the cells prior to PCR. We perform PCR from cell mixes either immediately after cell sorting (Figure 2), or from the whole dish without sorting (Figure 2—figure supplement 2).

Despite the differences in exact numbers, Suzuki et al. also reports ~ 40% INDELs at either one or both ends of insertion when their “2C” template is used (Extended Figure 2g in Suzuki et al.). This “2C” template is similar to our template except that it targets exons. As quoted above, the majority of transfected, but unlabeled, alleles contain INDELs, which can cause mutations at the mRNA level. At these INDEL rates, CRISPIE has significant advantages by significantly lowering the error rate at both the mRNA and protein levels (< 2% error rate across conditions; Figure 2 of our manuscript).

– Please report the data in Figures 3B-C as absolute labeling efficiencies instead of relative efficiencies. Normalizing efficiency to 1 is misleading and obscures the actual frequency of successful labeling.

As suggested, we have redone the experiment and we now provide the absolute labeling efficiency (new Experiment #5; Figure 3B and 3C). The new numbers are qualitatively similar to previous results, although the specific number is slightly different (e.g., the difference between intron and exon labeling is now > 5-fold instead of the previously reported 4-fold difference).